# Non-equilibrium steady state formation in 3+1 dimensions

Christian Ecker[1*], Johanna Erdmenger[2], Wilke van der Schee[3]

**1** Institut für Theoretische Physik, Goethe Universität, Max-von-Laue-Str. 1, 60438 Frankfurt am Main, Germany
**2** Institut für Theoretische Physik und Astrophysik, Julius-Maximilians-Universität Würzburg, Am Hubland, 97074 Würzburg, Germany
**3** Theoretical Physics Department, CERN, CH-1211 Genève 23, Switzerland
* ecker@itp.uni-frankfurt.de

April 9, 2021

## Abstract

We present the first holographic simulations of non-equilibrium steady state formation in strongly coupled $\mathcal{N} = 4$ SYM theory in 3+1 dimensions. We initially join together two thermal baths at different temperatures and chemical potentials and compare the subsequent evolution of the combined system to analytic solutions of the corresponding Riemann problem and to numeric solutions of ideal and viscous hydrodynamics. The time evolution of the energy density that we obtain holographically is consistent with the combination of a shock and a rarefaction wave: A shock wave moves towards the cold bath, and a smooth broadening wave towards the hot bath. Between the two waves emerges a steady state with constant temperature and flow velocity, both of which are accurately described by a shock+rarefaction wave solution of the Riemann problem. In the steady state region, a smooth crossover develops between two regions of different charge density. This is reminiscent of a contact discontinuity in the Riemann problem. We also obtain results for the entanglement entropy of regions crossed by shock and rarefaction waves and find both of them to closely follow the evolution of the energy density.

# 1 Introduction

Describing the far-from-equilibrium dynamics of strongly coupled quantum systems is extremely challenging. Gauge/gravity duality [1–3] provides important insights by mapping the dynamics of certain strongly coupled non-Abelian gauge theories to the dynamics of classical gravity in higher dimensions. This approach has been successfully applied to study the dynamics of the strongly coupled quark-gluon plasma in relativistic heavy ion collisions [4] and strongly correlated condensed matter systems [5,6]. The picture arises that strongly coupled far-form-equilibrium states typically evolve extremely fast towards a hydrodynamic regime before reaching a state of thermal equilibrium after a sufficiently long time. An important exception to this rule are quantum states on dynamical backgrounds that arise for example in the context of cosmology [7,8].

Another exception are systems driven by external sources that therefore never reach thermal equilibrium, but instead evolve towards a steady-state with non-vanishing fluxes, but time independent thermodynamic properties. An important example, which is the subject of this work, is a cold-hot interface of two identical copies of a quantum critical system at different temperatures and chemical potentials from which a Non-Equilibrium Steady State (NESS) emerges between two outgoing waves.

The properties and formation process of NESSs have been studied extensively using various different approaches. In 1+1 dimensional conformal field theories ($\text{CFT}_2$), the heat and charge flows of the system considered show universal behaviour [9–13]. The appearance of the NESS as well as its properties are insensitive to the details of the initial state and depend only on fundamental parameters, the central charge and current algebra level, of the $\text{CFT}_2$, as well as on the initial temperatures and chemical potentials of the two copies. More recently, exact results have also been obtained in $T\bar{T}$-deformed $\text{CFT}_2$ [14].

In [15] it was shown that the NESSs in holographic $\text{CFT}_2$ are dual to Lorentz boosted black brane geometries in the bulk. The Einstein equations for the gravity dual determine the geometry such that even far-from-equilibrium solutions such as propagating shockwaves are related by large coordinate transformations to static $\text{AdS}_3$. The holographic entanglement entropy has been studied for the two-dimensional case in [16] using the Hubeny-Rangamani-Takayanagi prescription [17,18].

One might ask if these NESSs are a curiosity of integrable $\text{CFT}_2$ or if they also exist in more general theories and dimensions higher than two? This question was addressed in several studies by constructing solutions of the Riemann problem in relativistic hydrodynamics [15,19–22], in holographic $\text{CFT}_3$ [23], in theories with gravity duals in the limit of large number of dimensions [24] and in non-relativistic theories with Lifshitz scale symmetry [25]. This led to the insight that the formation of NESSs does not rely on conformal symmetry or integrability, but rather is a universal feature of the hydrodynamic description of any fluid, independent of the underlying equation of state.

However, the details on how a NESS dynamically emerges from the interface depend crucially on the number of dimensions in which the system lives. In 1+1 dimensions, the NESS region emerges between two planar shockwaves travelling at the speed of light outwards from the interface. In higher dimensions however, this is not the case any more: For entropic reasons, the wave front moving towards the hot side is a smoothly broadening rarefaction wave, while the wave front moving towards the cold side is still given by a shockwave [20,21].

One goal of this work is to sharpen the picture of the formation process of NESSs at

strong coupling in four spacetime dimensions. First, we compare the evolution of both the stress tensor and the charge density in the strongly coupled field theory to analytic solutions and numeric approximations of the Riemann problem in ideal and viscous hydrodynamics, respectively. We confirm that the double-shock solution violates the second law of thermodynamics also in 3+1 dimensions. Also, as we discuss both for the case of two shock waves as well as for the shock+rarefaction wave combination, the charge density displays a discontinuity within the NESS region, for which we plot examples. Then, moving on to the gravity dual in five dimensions, we numerically evolve the dual gravity problem to obtain the fully far-from-equilibrium quantum dynamics in principle also beyond the hydrodynamic regime. In particular, we numerically establish and analyse the gravity dual of the rarefaction wave. While being entirely smooth, our holographic solutions agree to very good numerical accuracy with the shock+rarefaction wave scenario. Importantly, our results are in line with the proposal of [20] that a NESS forms also for the shock+rarefaction case, i.e. the spreading of the rarefaction wave is not so large as to impede the NESS formation. We present a quantitative study of the deviations between the shock+shock and shock+rarefaction cases, the hydrodynamic simulation and our holographic solutions. We find them to be generically small. Nevertheless, our holographic solutions favour the shock+rarefaction scenario.

A further goal is to generalise the holographic entanglement entropy (HEE) calculation of [16] to four spacetime dimensions. In [16], where two 1+1-dimensional shockwaves and their gravity dual were considered, the time dependence of the HEE for a strip entangling region was shown to display universal behaviour and to satisfy a velocity bound. In contrast to $CFT_2$, where the dual Riemann problem has a closed solution [26], in dimensions larger than two it is necessary to solve the extremal surface problem for the HEE numerically [27]. We perform a numerical analysis of the HEE for infinite strip regions of different width. In particular for the shock+rarefaction case, we find that a convenient way to make physical statements about the HEE time evolution is to compare it to the time evolution of the energy density. We compare the time evolution of both HEE and energy density during the passing of the shock and rarefaction waves. We find the time evolution of HEE and energy density to be very similar, the main difference being that the HEE trails the energy density by a small amount. This effect is more pronounced in the rarefaction case, where the wave takes a relatively long time to move through the entangling region.

The paper is structured as follows. Sec. 2 is a review of the Riemann problem in ideal hydrodynamics. In particular, we recall the derivation of analytic solutions with two shock waves and solutions with one shock and one rarefaction wave. In Sec. 3 we introduce the holographic model which is Einstein-Maxwell gravity in five dimensions. In Sec. 4 we discuss our setup of the HEE computation. In Sec. 5 we present the time evolution of the stress tensor and charge density obtained from the holographic model and compare them to analytic and numeric solutions of the corresponding Riemann problem in ideal and viscous hydrodynamics. We then analyse the holographic entanglement entropy for shock and rarefaction waves. In Sec. 6 we conclude and point towards a number of interesting future directions. In two appendices we derive the Rankine-Hugoniot jump conditions and provide numeric evidence that our results are independent on how we approximate the initial interface of the Riemann problem on the gravity side.

## 2   Riemann problem in ideal hydrodynamics

Prior to discussing the holographic calculation, it is useful to review the Riemann problem in ideal hydrodynamics in presence of a conserved U(1) charge. We start by defining the stress tensor and charge current of a relativistic fluid,

$$T^{\mu\nu} = (\mathcal{E} + \mathcal{P})u^\mu u^\nu + \mathcal{P}\eta^{\mu\nu}, \quad J^\mu = nu^\mu, \tag{1}$$

where $\mathcal{E}$, $\mathcal{P}$, $n$ and $\eta$ denote energy density, pressure, charge density and the mostly plus Minkowski metric, respectively. We are interested in one dimensional relativistic flows for which the normalised velocity can be written as

$$u^\mu = \gamma(1, v, \vec{0}), \quad u^2 = u^\mu u^\nu \eta_{\mu\nu} = -1. \tag{2}$$

The Lorentz factor $\gamma = 1/\sqrt{1 - v^2}$ is expressed in terms of the local fluid velocity $v$. The equations of motion of the fluid are the conservation laws for the stress tensor and the charge current

$$\partial_\mu T^{\mu\nu} = 0, \quad \partial_\mu J^\mu = 0. \tag{3}$$

These equations need to be closed by an equation of state (EoS) relating pressure to energy and charge density. In what follows we neglect the dependence of the pressure on the charge density and assume the fluid to be conformally invariant,

$$\mathcal{P}(\mathcal{E}) = c_s^2 \mathcal{E}, \quad c_s^2 = \frac{1}{d}, \tag{4}$$

where $c_s$ is the speed of sound and $d$ the number of spatial dimensions. We can now state the Riemann problem which is an initial value problem for (3) with piecewise constant initial conditions at time $t = 0$ for the energy and charge density with a planar discontinuity at $x = 0$,

$$\mathcal{E}(0, x) = \begin{cases} \mathcal{E}_C & \forall x < 0 \\ \mathcal{E}_H & \forall x > 0 \end{cases}, \quad n(0, x) = \begin{cases} n_C & \forall x < 0 \\ n_H & \forall x > 0 \end{cases}. \tag{5}$$

In the following we assume, without loss of generality, $\mathcal{E}_C < \mathcal{E}_H$, where subscripts $C$ and $H$ denote the "cold" and the "hot" side of the system, respectively.

### 2.1   Double shock solution

One possible solution of the Riemann problem consists of two shock discontinuities moving in opposite directions. In this case, there are three different regions, for which the stress tensor is given by

$$T_C^{\mu\nu} = \begin{pmatrix} \mathcal{E}_C & 0 \\ 0 & c_s^2 \mathcal{E}_C \end{pmatrix}, \quad T_S^{\mu\nu} = c_s^2 \mathcal{E}_S \left( (1 + 1/c_s^2) u^\mu u^\nu + \eta^{\mu\nu} \right), \quad T_H^{\mu\nu} = \begin{pmatrix} \mathcal{E}_H & 0 \\ 0 & c_s^2 \mathcal{E}_H \end{pmatrix}, \tag{6}$$

respectively. In this subsection we suppress the $d - 1$ transverse coordinates for clarity. The middle region, labelled by subscript $S$ for steady state, is described by a fluid with local restframe energy density $\mathcal{E}_S$ moving with velocity $v_S$,

$$u^\mu = \gamma(1, v_S), \quad \gamma = 1/\sqrt{1 - v_S^2}. \tag{7}$$

The charge density can in addition develop a so-called contact discontinuity in the central region where the pressure and the velocity are continuous. This means that the solution for the charge density consists of four different regions in general, with local charge densities of $n_C$, $n_1$, $n_2$ and $n_H$ such that

$$J^\mu_C = n_C \begin{pmatrix} 1 \\ 0 \end{pmatrix}, \quad J^\mu_1 = n_1 \gamma \begin{pmatrix} 1 \\ v_S \end{pmatrix}, \quad J^\mu_2 = n_2 \gamma \begin{pmatrix} 1 \\ v_S \end{pmatrix}, \quad J^\mu_H = n_H \begin{pmatrix} 1 \\ 0 \end{pmatrix}. \tag{8}$$

Energy-momentum and charge conservation then imply the Rankine-Hugoniot jump conditions (for details see Appendix A) for the stress tensor at the left and right moving shock,

$$\begin{aligned}
v_C(T^{tt}_C - T^{tt}_S) &= T^{xt}_C - T^{xt}_S & v_C(T^{tx}_C - T^{tx}_S) &= T^{xx}_C - T^{xx}_S \\
v_H(T^{tt}_S - T^{tt}_H) &= T^{xt}_S - T^{xt}_H & v_H(T^{tx}_S - T^{tx}_H) &= T^{xx}_S - T^{xx}_H \,,
\end{aligned} \tag{9}$$

and correspondingly for the charge densities

$$v_C(J^t_C - J^t_1) = J^x_C - J^x_1 \,, \qquad v_H(J^t_2 - J^t_H) = J^x_2 - J^x_H \,. \tag{10}$$

These conditions determine the charge densities $n_1$ and $n_2$, the shock velocities $v_C$ and $v_H$, as well as the boost velocity $v_S$ and energy density $\mathcal{E}_S$ of the NESS region in terms of the boundary conditions $n_C$, $n_H$, $\mathcal{E}_C$ and $\mathcal{E}_H$,

$$v_S = -\frac{c_s(1-\chi)}{\sqrt{(1+c_s^2\chi)(c_s^2+\chi)}} \qquad v_C = -c_s\sqrt{\frac{1+c_s^2\chi}{c_s^2+\chi}} \qquad v_H = c_s\sqrt{\frac{c_s^2+\chi}{1+c_s^2\chi}}$$

$$\mathcal{E}_S = \sqrt{\mathcal{E}_C\mathcal{E}_H} \qquad\qquad n_1 = n_C\sqrt{\frac{1+c_s^2\chi}{\chi(c_s^2+\chi)}} \qquad n_2 = n_H\sqrt{\frac{\chi(c_s^2+\chi)}{1+c_s^2\chi}} \,, \tag{11}$$

where $\chi = \sqrt{\mathcal{E}_C/\mathcal{E}_H}$. In Fig. 1 we plot the energy and charge density for different values of $\chi$ for $d = 3$. In Fig. 2 (left) we show the velocities as a function of $\chi$ for $d = 3$, whereby we note that $1 > v_H > c_s$ and $c_s > v_C$. We express the solutions as functions of the ratio of the $x$-coordinate in which the shocks propagate and time $t$, $x/t$. Note that on the right of both plots, the value of the quantities shown is one since $\mathcal{E} = \mathcal{E}_H$. For special ratios of the initial charge and energy densities

$$\frac{n_C}{n_H} = \frac{\chi(c_s^2+\chi)}{1+c_s^2\chi} \,, \tag{12}$$

the contact discontinuity of the charge is absent, implying $n_1 = n_2$. For $d = 3$ spatial dimensions and $n_C/n_H = 1/2$, this is the case for $\chi = (\sqrt{73}-1)/12$, as also shown in Fig. 1. For larger values of $\chi$ the charge density becomes non-monotonic: the difference in the energy densities generates such a strong flow of charge that charge builds up next to the cold bath.

As noted in [20, 24], a shock wave moving into a region of higher energy density and pressure violates the entropy condition

$$\partial_\mu s^\mu \geq 0 \,, \tag{13}$$

where $s^\mu = \frac{\mathcal{E}+\mathcal{P}}{T} u^\mu = k\mathcal{E}^{\frac{1}{1+c_s^2}} u^\mu$ is the entropy density current and $k$ a constant that depends on the microscopic properties of the theory. Using the entropy currents in the left, central and right regions

$$s^\mu_C = k\mathcal{E}^{\frac{1}{1+c_s^2}}_C \begin{pmatrix} 1 \\ 0 \end{pmatrix}, \quad s^\mu_S = k\mathcal{E}^{\frac{1}{1+c_s^2}}_S \gamma \begin{pmatrix} 1 \\ v_S \end{pmatrix}, \quad s^\mu_H = k\mathcal{E}^{\frac{1}{1+c_s^2}}_H \begin{pmatrix} 1 \\ 0 \end{pmatrix}, \tag{14}$$

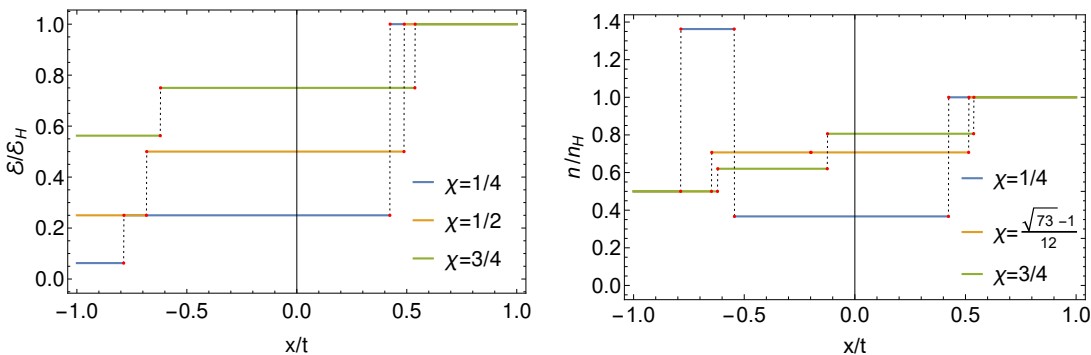

Figure 1: Energy density (left) and charge density (right) of the double shock solution. The energy plot on the left shows the two heat baths and the NESS region. The charge plot displays the additional contact discontinuity that is absent for the middle value of $\chi$ given.

we may evaluate the corresponding jump conditions for the entropy current across the shock waves,

$$\Delta s_C = v_C \left(s_C^t - s_S^t\right) - \left(s_C^x - s_S^x\right) = k c_s \mathcal{E}_H^{\frac{1}{1+c_s^2}} \left(\chi^{\frac{3}{2(1+c_s^2)}} - \chi^{\frac{2}{1+c_s^2}} \sqrt{\frac{1 + c_s^2 \chi}{c_s^2 + \chi}}\right), \qquad (15)$$

$$\Delta s_H = v_H \left(s_S^t - s_H^t\right) - \left(s_S^x - s_H^x\right) = k c_s \mathcal{E}_H^{\frac{1}{1+c_s^2}} \left(\chi^{\frac{1-c_s^2}{2(1+c_s^2)}} - \sqrt{\frac{c_s^2 + \chi}{1 + c_s^2 \chi}}\right). \qquad (16)$$

For the number of spatial dimensions $d > 1$, these expressions reveal that both shocks violate the jump condition for the entropy current, i.e. $\Delta s_{C/H} \neq 0$. However, as shown in Fig. 2 (right), the right-moving shock (orange curve) gives $\Delta s_H < 0$ and therefore violates also the entropy condition (13) and is in conflict with the second law of thermodynamics. The entropy

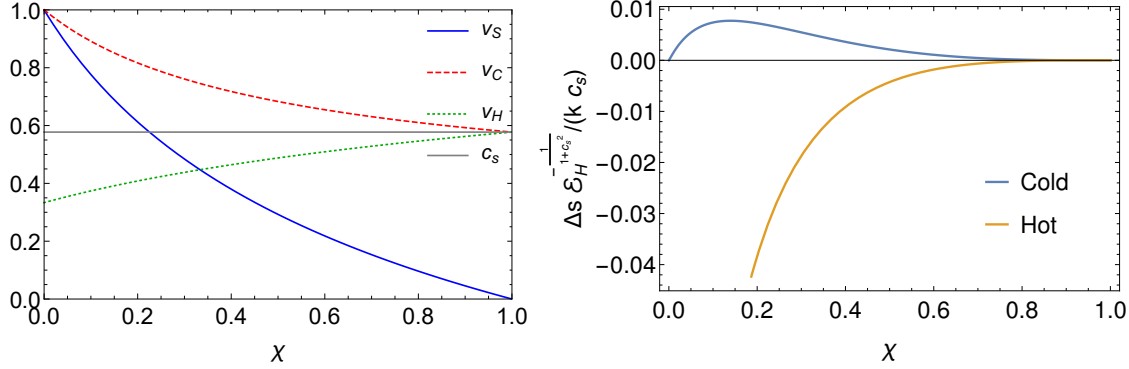

Figure 2: (left) The velocities of the steady state flow $(v_S)$, and the left- and right-moving shock velocities $v_C$ and $v_H$ in comparison to the sound velocity $c_s$ for $d = 3$ according to Eqn. (11). (right) Change in entropy across the left and right moving shock waves, as function of $\chi = \sqrt{\mathcal{E}_C/\mathcal{E}_H}$. 'Cold' refers to the left-moving and 'hot' to the right-moving shock. The normalization factor is motivated by (15).

flow across the right-moving shock is negative and monotonous. In the limit $\chi \to 0$, it is bounded by

$$\lim_{\chi \to 0} \Delta s_H = -k \, c_s^2 \left(\mathcal{E}_H\right)^{\frac{1}{1+c_s^2}}. \qquad (17)$$

The flow across the left-moving shock is completely suppressed in this limit , i.e. $\lim_{\chi \to 0} \Delta s_C = 0$. Interestingly, for each dimension $d$ the entropy flow across the left mover has a local maximum at some $\chi_d^*$ for which the shock produces a maximum amount of entropy. For $d = 2, 3, 4$, the corresponding value of $\chi_d$ is given by

$$\chi_2^* \approx 0.1301 \,, \quad \chi_3^* \approx 0.1397 \,, \quad \chi_4^* \approx 0.1428 \,, \quad \dots . \tag{18}$$

## 2.2 Shock + rarefaction wave solution

A simple way to obtain a physical solution is to replace the unphysical shock wave with a so-called rarefaction wave [20]. A rarefaction wave is a smooth, self-similar solution that by construction saturates the entropy condition (13) and depends on $x$ and $t$ only via $\xi = x/t$. In the $\xi$ coordinate, the conservation equations for the stress tensor become

$$\xi \frac{\mathrm{d}}{\mathrm{d}\xi} \left( \mathcal{E} \frac{1 + c_s^2 v^2}{1 - v^2} \right) = \frac{\mathrm{d}}{\mathrm{d}\xi} \left( \mathcal{E} v \frac{1 + c_s^2}{1 - v^2} \right) \,, \qquad \xi \frac{\mathrm{d}}{\mathrm{d}\xi} \left( \mathcal{E} v \frac{1 + c_s^2}{1 - v^2} \right) = \frac{\mathrm{d}}{\mathrm{d}\xi} \left( \mathcal{E} \frac{c_s^2 + v^2}{1 - v^2} \right) \,, \tag{19}$$

which can be rearranged to

$$\begin{pmatrix} 0 \\ 0 \end{pmatrix} = M \begin{pmatrix} \frac{\mathrm{d}}{\mathrm{d}\xi} \mathcal{E} \\ \frac{\mathrm{d}}{\mathrm{d}\xi} v \end{pmatrix} \,. \tag{20}$$

This system of ordinary differential equations has solutions different from $\mathcal{E} = v = 0$ if and only if

$$\det M = \left( c_s^2 \xi^2 - 1 \right) v(\xi)^2 - 2 \left( c_s^2 - 1 \right) \xi v(\xi) + c_s^2 - \xi^2 = 0 \,, \tag{21}$$

which gives a relation for the local velocity in terms of $\xi$,

$$v(\xi) = \frac{\xi \pm c_s}{1 \pm c_s \xi} \,, \tag{22}$$

where the plus (minus) sign corresponds to a left (right) moving wave. Next we demand local entropy conservation $\partial_\mu s^\mu = 0$ for the right-moving wave,

$$\xi \frac{\mathrm{d}}{\mathrm{d}\xi} \left( \frac{\mathcal{E}^{\frac{1}{1+c_s^2}}}{\sqrt{1 - v^2}} \right) = \frac{\mathrm{d}}{\mathrm{d}\xi} \left( \frac{v \mathcal{E}^{\frac{1}{1+c_s^2}}}{\sqrt{1 - v^2}} \right) \,. \tag{23}$$

From this we can express the energy density of the rarefaction wave as

$$\mathcal{E} = \mathcal{E}_H \left( \frac{(c_s - 1)(\xi + 1)}{(c_s + 1)(\xi - 1)} \right)^{\frac{1 - c_s^2}{2 c_s}} \,, \tag{24}$$

where we fixed the integration constant by $\mathcal{E}(\xi = c_s) = \mathcal{E}_H$. Similarly the conservation law for the charge current

$$\xi \frac{\mathrm{d}}{\mathrm{d}\xi} \left( \frac{n}{\sqrt{1 - v^2}} \right) = \frac{\mathrm{d}}{\mathrm{d}\xi} \left( \frac{nv}{\sqrt{1 - v^2}} \right) \,, \tag{25}$$

can be solved for the charge density in the rarefaction region

$$n(\xi) = n_H \left( \frac{(1 + \xi)(1 - c_s)}{(1 - \xi)(1 + c_s)} \right)^{c_s/2} \,, \tag{26}$$

where we used (22) to express the local velocity and $n(\xi = c_s) = n_H$ to fix the integration constant. By combining (22) and (24) we can express the energy density in the central region in terms of the flow velocity

$$\mathcal{E}_S = \mathcal{E}_H \left( \frac{1 + v_S}{1 - v_S} \right)^{\frac{1 - c_s^2}{2c_s}}. \tag{27}$$

A similar expression can be derived from the Rankine-Hugoniot jump conditions (9) for the left moving shock wave

$$v_C \left( \mathcal{E}_C - \mathcal{E}_S \frac{1 + c_s^2 v_S^2}{1 - v_S^2} \right) = -\frac{\mathcal{E}_S(1 + c_s^2)v_S}{1 - v_S^2}, \quad -v_C \frac{\mathcal{E}_S(1 + c_s^2)v_S}{1 - v_S^2} = c_s^2 \mathcal{E}_C - \frac{\mathcal{E}_S(v_S^2 + c_s^2)}{1 - v_S^2}, \tag{28}$$

from which we obtain

$$\mathcal{E}_S = \mathcal{E}_C \frac{2c_s^2 + v_S^2 + c_s^4 v_S^2 \pm v_S(1 + c_s^2)\sqrt{4c_s^2 + (c_s^2 - 1)^2 v_S^2}}{2c_s^2(1 - v_S^2)}, \tag{29}$$

$$v_C = \frac{v_S^2 + c_s^2 \left( 1 - (1 - v_S^2) \left( \frac{1 - v_S}{1 + v_S} \right)^{\frac{1 + c_s^2}{2c_s}} \chi^2 \right)}{v_S(1 + c_s^2)}, \tag{30}$$

where the solution with the minus (plus) sign corresponds to a right (left) moving rarefaction wave. Combining (27) and (29) fixes a unique value for $v_S$ which we are only able to determine numerically. With this in mind, we express the charge densities in terms of $v_S$,

$$n_1 = n_C \frac{(v_S^2 - 1)\chi^2 \left( \frac{1 - v_S}{1 + v_S} \right)^{\frac{c_s^2 + 1}{2c_s}} + 1 + \frac{v_S^2}{c_s^2}}{\sqrt{1 - v_S^2} \left( 1 - \chi^2 \left( \frac{1 + v_S}{1 + v_S} \right)^{\frac{c_s^2 + 1}{2c_s}} \right)}, \quad n_2 = n_H \left( \frac{(1 + v_S)(1 - c_s)}{(1 - v_S)(1 + c_s)} \right)^{c_s/2}. \tag{31}$$

In Fig. 3 we plot some examples for energy and charge density. In contrast to the shock+shock case the rarefaction wave provides a continuous solution near the hot bath. From the figure it seems that $\chi = (\sqrt{73} - 1)/12$ again provides a solution without a contact discontinuity, but in fact a careful numerical comparison shows that $n_1 - n_2 \approx 0.001092$. We also note that the direction the rarefaction wave travels is solely determined by the presence of the hot bath, and does not depend on $n_H$ being higher or smaller than $n_C$, which is clear from Fig. 3 (right). Similarly, the contact discontinuity does not get replaced by a rarefaction wave, as in the steady state rest frame it is just a connection between two baths of different charge densities. In practice charge will diffuse from the higher to the lower charge density, but this is a process that is parametrically slower than the shock and rarefaction waves (see also Section 5). All these solutions are now potential physical solutions that satisfy the second law. In the remainder of this work, we will investigate these further from a microscopic perspective.

## 3 Riemann problem in holography

### 3.1 Holographic model

The holographic dual model that we use is five-dimensional Einstein-Maxwell gravity with negative cosmological constant. This allows us to study the dynamics of the stress tensor and

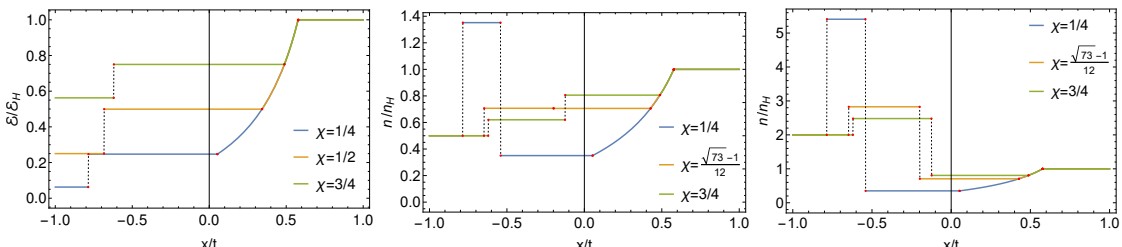

Figure 3: Energy density (left) and charge density (middle and right) for the shock+rarefaction wave solution. The rarefaction wave moving to the hot bath appears on the right. This figure is to be compared to the two-shock solution displayed in Fig. 1.

a conserved $U(1)$ current in the dual field theory. The action of the gravity system is given by

$$S = \frac{1}{16\pi G_N} \int_{\mathcal{M}} \mathrm{d}^5 x \sqrt{-g} \left( R + \frac{12}{L^2} - \frac{e^2 L^2}{4} F_{MN} F^{MN} \right) + \frac{1}{8\pi G_N} \int_{\partial \mathcal{M}_\epsilon} \mathrm{d}^4 x \sqrt{-\gamma} K + S_{\mathrm{ct}} \,, \quad (32)$$

where $G_N$ is Newton's constant, $L$ is the asymptotic AdS radius, $R$ is the Ricci scalar of the bulk geometry on a manifold $\mathcal{M}$ with flat boundary $\partial \mathcal{M}$ and bulk metric $g_{MN}$, $F_{MN} \equiv \partial_{[M} A_{N]}$ is the electromagnetic field strength with $A_M$ the U(1) gauge field in the bulk and the coupling constant $e$ controls the strength of the electromagnetic field. The trace of the extrinsic curvature $K$ of the induced metric $\gamma_{\mu\nu}$ and the counter-term [28, 29] are to be evaluated at a radial slice $\partial \mathcal{M}_\epsilon$ close to the boundary and are necessary to render the variational principle well-defined and the on-shell action finite.

The action (32) can be viewed as a consistent truncation of the dimensional reduction of type IIB supergravity on $S^5$. In this case the dual gauge theory is $\mathcal{N} = 4$ Super Yang-Mills and the $U(1)$ current arises from the R-symmetry of this theory.[1] In the context of this work we see (32) simply as bottom-up model that incorporates the dynamics of the stress tensor and a conserved $U(1)$ current in the dual gauge theory.

The equations of motion that follow from (32) are

$$R_{MN} + \frac{4}{L^2} g_{MN} = \frac{e^2 L^2}{2} \left( F_{MP} F_N{}^P - \frac{1}{6} g_{MN} F^2 \right) \,, \quad (33a)$$

$$\nabla^M F_{MN} = 0 \,. \quad (33b)$$

The ground state of the theory is given by a constant gauge field configuration on AdS$_5$,

$$ds^2 = \frac{L^2}{u^2} \left( \mathrm{d}u^2 + \eta_{\mu\nu} \mathrm{d}x^\mu \mathrm{d}x^\nu \right) \,, \quad A_M = \mathrm{const.} \quad (34)$$

A general solution of the Maxwell equations near the AdS boundary takes in axial gauge $(A_u = 0)$ the form

$$A_\mu(u, x^\mu) = a_\mu^{(0)}(x^\mu) + u^2 \left( a_\mu^{(1)}(x^\mu) + \tilde{a}_\mu^{(1)}(x^\mu) \log u \right) + \dots \,, \quad (35)$$

---

[1]The full five-dimensional action for this truncation would include a Chern-Simons term (see e.g. [30]), but this will play no role in our analysis and we have therefore omitted it.

where the coefficient $a_\mu^{(0)}(x^\mu)$ is identified as coupling of the global U(1) current $J^\mu(x^\mu)$ in the quantum field theory. A constant $a_\mu^{(1)}(x^\mu)$ is then, by the holographic dictionary (see e.g. [31]), identified as chemical potential $\mu$ for a global charge density $\rho$

$$\mu \equiv a_t^{(0)}, \quad \langle J^t \rangle = \rho \equiv -\frac{e^2 L^4}{8\pi G_N} a_t^{(1)}. \tag{36}$$

In the following we will be interested in solutions dual to field theory states in the grand canonical ensemble, i.e., thermodynamic states characterised by fixed chemical potential and temperature. Such states are dual to Reissner–Nordström (RN) black branes,

$$ds_{\text{RN}}^2 = \frac{L^2}{u^2}\left(-f(u)\mathrm{d}t^2 + f(u)^{-1}\mathrm{d}u^2 + \mathrm{d}\vec{x}^2\right), \quad A_t(u) = \mu\left(1 - \frac{u^2}{u_{\text{h}}^2}\right) \tag{37a}$$

$$f(u) = 1 - M\frac{u^4}{u_{\text{h}}^4} + Q^2\frac{u^6}{u_{\text{h}}^6}, \quad M = 1 + Q^2, \quad Q^2 = \frac{\mu^2 u_{\text{h}}^2 e^2}{3L^2}, \tag{37b}$$

where $u = u_{\text{h}}$ is the radial location of the horizon defined by $f(u_{\text{h}}) = 0$ and $f(0) = 1$ fixes the boundary metric to Minkowski. The temperature of the field theory state dual to the geometry (37) is given by the Hawking temperature of the horizon. It can be derived by demanding periodicity $\beta$ of time circles in the Euclidean continuation of the line element,

$$T = \frac{1}{\beta} = -\frac{f'(u_{\text{h}})}{4\pi} = \frac{Q^2 - 2}{2\pi u_{\text{h}}}. \tag{38}$$

By the Bekenstein–Hawking formula [32,33], the entropy density is proportional to the horizon area,

$$s = \frac{L^3}{4G\, u_{\text{h}}^3}. \tag{39}$$

The charge density then follows from using (37a) in (36),

$$\rho = \frac{e^2 L^2 \mu}{8\pi G_N u_{\text{h}}^2}. \tag{40}$$

RN-geometries (37) with different $T$ and $\mu$ will serve as initial conditions to the left and to the right of the interface in the Riemann problem and at the same time provide the necessary boundary conditions at spatial infinity to solve the initial value problem. In the next section we explain that NESSs emerging from the interface are dual to Lorentz transformed (boosted) versions of (37) and present the method we use to simulate their formation.

## 3.2 Holographic steady states

The holographic duality maps NESSs in the field theory to boosted black brane geometries on the gravity side [15],

$$ds_{\text{NESS}}^2 = \frac{L^2}{u^2}\left[\mathrm{d}u^2 f(u)^{-1} - f(u)\left(\mathrm{d}t\cosh\eta - \mathrm{d}z\sinh\eta\right)^2 + \left(\mathrm{d}z\cosh\eta - \mathrm{d}t\sinh\eta\right)^2 + \mathrm{d}\mathbf{x}_\perp^2\right], \tag{41}$$

where the rapidity $\eta$ is related to the fluid velocity $v_s$ in the steady state by $\eta = \tanh^{-1} v_s$. At this point it is important to emphasise that explicit relations between the temperature, fluid

velocity, etc. of the holographic steady state and the properties of the hot and cold reservoirs are only known in $d = 2$ [20]. The temperatures of the cold and the hot reservoir ($T_{C,H}$) depend on the corresponding chemical potentials by (38) ($\mu_{C,H}$) and radial horizon positions ($u_{\mathrm{h}(C,H)}$),

$$T_{C,H} = \frac{Q_{C,H}^2 - 2}{2\pi u_{\mathrm{h}(C,H)}} = \frac{\mu_{C,H}^2 u_{\mathrm{h}(C,H)}}{6\pi^2}\frac{e^2}{L^2} - \frac{1}{2\pi u_{\mathrm{h}(C,H)}}\,. \tag{42}$$

The dual geometry for the steady state regime can be approximated as

$$ds^2 = \begin{cases} ds_{\mathrm{RN,C}}^2 & \text{if} \quad z \lesssim v_C t\,, \\ ds_{\mathrm{NESS}}^2 & \text{if} \quad v_C t \lesssim z \lesssim v_H t\,, \\ ds_{\mathrm{RN,H}}^2 & \text{if} \quad z \gtrsim v_H t\,, \end{cases} \tag{43}$$

where $ds_{\mathrm{RN,C}}^2$ ($ds_{\mathrm{RN,H}}^2$) is (37) for $T = T_C(T_H)$ and $\mu = \mu_C(\mu_H)$. In the hydrodynamic limit, the left ($v_C$) and right ($v_H$) moving wave velocities are well approximated by (28) with (22) evaluated for $\xi = c_s$. It is important to note that if $\chi$ is not close to unity we expect more complicated solutions that in particular include the rarefaction waves introduced in Sec. 2.2. Similarly we can arrive at approximate expressions for the gauge field using the formulae in Sec. 2.2. This is only an approximate solution, because the precise form of the metric in the vicinity of the left and right moving waves is not available in closed form. In Sec. 3 we present numeric evidence that the expressions for temperature, fluid velocity and charge density derived for the shock+rarefaction solution in Sec. 2 are, at sufficiently late time, in excellent agreement with the results of the holographic simulation. In contrast to Sec. 2, where we denote the spatial coordinate in which the waves propagate by $x$, we denote the corresponding coordinate here and in the following by $z$.

To study the Riemann problem in full detail we construct the solution numerically. This will allow us to analyse the precise shapes of the propagating waves, how they change in time and how they compare to the analytically constructed shock and rarefaction wave solution in ideal hydrodynamics. For this we make the simplifying assumption that the strength of the electromagnetic field is small, i.e. $e\,L \ll 1$. In this limit the backreaction of the gauge field to the metric is subleading and the right hand side of (33a) vanishes. This means our charged results are leading order results in a small $e\,L$ expansion.

For the metric and the gauge field we follow [34–37] and use the ansätze

$$ds_{\mathrm{EF}}^2 = -C\mathrm{d}t^2 + 2\mathrm{d}r\mathrm{d}t + 2G\mathrm{d}z\mathrm{d}t + S^2\left(e^B\mathrm{d}\mathbf{x}_\perp^2 + e^{-2B}\mathrm{d}z^2\right)\,, \tag{44a}$$

$$A_M = A_t\mathrm{d}t + A_z\mathrm{d}z\,, \tag{44b}$$

where all functions depend on the Eddington–Finkelstein like time coordinate $t$, the longitudinal coordinate $z$ and the AdS bulk coordinate $r$, but not on the two transverse coordinates $\mathbf{x}_\perp$. The explicit form of the corresponding equations of motion for the metric can for example be found in [34, 38]. The Maxwell equations with (44b) as ansatz for the gauge field can be

written as

$$S^3 F'_{rt} = - e^{2B} \left( F_{rz} \left( S \left( 2\tilde{B} - G' \right) + \tilde{S} \right) + S\tilde{F}_{rz} \right) - 3S^2 F_{rt} S' \,, \tag{45a}$$

$$\begin{aligned} S^3 F'_{tz} = &\frac{1}{4} S^2 \big( - 2S(\tilde{F}_{rt} + F_{rz} \left( CB' + 2\dot{B} + C' \right) + 2B' \left( GF_{rt} + F_{tz} \right) \\ &+ CF'_{rz} + F_{rt}G' ) + S' \left( -CF_{rz} + 10GF_{rt} - 2F_{tz} \right) - 2\dot{S}F_{rz} \big) \\ &+ e^{2B} G \left( F_{rz} \left( S \left( 2\tilde{B} - G' \right) + \tilde{S} \right) + S\tilde{F}_{rz} \right) \,, \end{aligned} \tag{45b}$$

$$\begin{aligned} 4S\dot{F}_{rz} = &- 2S \left( -\tilde{F}_{rt} + F_{rz} \left( CB' + 2\dot{B} + C' \right) + 2B' \left( GF_{rt} + F_{tz} \right) + F_{rt}G' \right) \\ &- S' \left( CF_{rz} + 2GF_{rt} + 2F_{tz} \right) - 2\dot{S}F_{rz} \,, \end{aligned} \tag{45c}$$

$$\begin{aligned} 2S^3 \dot{F}_{rt} = &e^{2B} \big( S \left( 2G \left( 2\tilde{B}F_{rt} + \tilde{F}_{rt} \right) + 4\tilde{B}F_{tz} + C\tilde{F}_{rz} + 2\tilde{G}F_{rt} + 2\tilde{F}_{tz} \right) + \\ &F_{rz} \left( 2S \left( C\tilde{B} + \tilde{C} + \dot{G} \right) + C\tilde{S} \right) + 2\tilde{S} \left( GF_{rt} + F_{tz} \right) \big) - 6S^2 \dot{S}F_{rt} \,, \end{aligned} \tag{45d}$$

where $\dot{h} = \partial_t h + \frac{1}{2}Ch'$ and $\tilde{h} = \partial_z h - Gh'$. Once $F_{rz}$ is specified the first three equations can be used to respectively solve for $F_{rt}$, $F_{tz}$ and $\dot{F}_{rz}$, after which it is possible to obtain the time derivative of $F_{rz}$. The last equation is a constraint equation and can be used to monitor the accuracy of the numerical evolution. Close to the boundary, the solution for the metric and the gauge field can be expressed as power series in the radial coordinate,

$$C(r,\, t,\, z) = (r + \alpha)^2 - 2\partial_t \alpha + \frac{c_4}{r^2} + \frac{\partial_t c_4 - 4\alpha c_4}{2r^3} + O\left(r^{-4}\right), \tag{46a}$$

$$B(r,\, t,\, z) = \frac{b_4}{r^4} + \frac{15\partial_t b_4 + 2\partial_z f_4 - 60\alpha b_4}{15r^5} + O\left(r^{-6}\right), \tag{46b}$$

$$S(r,\, t,\, z) = r + \alpha - \frac{4\partial_z g_4 + 3\partial_t c_4}{60r^4} + O(r^{-5}), \tag{46c}$$

$$G(r,\, t,\, z) = \partial_z \alpha + \frac{g_4}{r^2} + \frac{4\partial_t g_4 + \partial_z c_4 - 10\alpha g_4}{5r^3} + O(r^{-4}), \tag{46d}$$

$$A_t(r,\, t,\, z) = \frac{a_{t,2}}{r^2} + \frac{\frac{2}{3}\partial_z a_{z,2} - 2\alpha a_{t,2}}{r^3} + O(r^{-4}), \tag{46e}$$

$$A_z(r,\, t,\, z) = \frac{a_{z,2}}{r^2} + \frac{-6\alpha a_{z,2} - \partial_z a_{t,2} + 3\partial_t a_{z,2}}{3r^3} + O(r^{-4}), \tag{46f}$$

where the function $\alpha(t, z)$ is a residual gauge freedom of the ansatz (44a) which we used to fix the horizon at $r = 1$. The functions $c_4(t, z)$, $b_4(t, z)$, $g_4(t, z)$, $a_{t,2}(t, z)$ and $a_{z,2}(t, z)$ are not determined by the near boundary analysis, but need to be extracted from a full bulk solution. The charge density and the holographic stress tensor in the field theory are then given by [36, 38, 39]

$$\langle J^\mu \rangle = \frac{e^2 L^4}{4\pi G_N} \begin{pmatrix} \bar{\rho} \\ \bar{\sigma} \\ 0 \\ 0 \end{pmatrix} \,, \quad \langle T^{\mu\nu} \rangle = \frac{1}{4\pi G_N} \begin{pmatrix} \mathcal{E} & \mathcal{S} & 0 & 0 \\ \mathcal{S} & \mathcal{P}_\parallel & 0 & 0 \\ 0 & 0 & \mathcal{P}_\perp & 0 \\ 0 & 0 & 0 & \mathcal{P}_\perp \end{pmatrix} \,, \tag{47}$$

where we defined the reduced variables for charge density ($\bar{\rho}$), charge flow ($\bar{\sigma}$), energy density ($\mathcal{E}$), pressure in longitudinal ($\mathcal{P}_\parallel$) and transverse ($\mathcal{P}_\perp$) directions and momentum flux ($\mathcal{S}$) [2].

---

[2] We note that in this section we switched to the $z$-coordinate to describe the boundary direction, to make

These quantities are related to the expansion coefficients as follows

$$\overline{\rho} = \frac{1}{2}a_{t,2}\,, \quad \overline{\sigma} = \frac{1}{2}a_{z,2}\,, \quad \mathcal{E} = -\frac{3}{4}a_4\,, \quad \mathcal{P}_{\parallel} = -\frac{1}{4}a_4 - 2b_4\,, \quad \mathcal{P}_{\perp} = -\frac{1}{4}a_4 + b_4\,, \quad \mathcal{S} = -f_4\,.$$
(48)

For $\mathcal{N} = 4$ SU($N_c$) SYM we have $G_N = \pi/2N_c^2$.

For the metric ansatz (44a) consistent initial conditions can be obtained by specifying $B(r, 0, z)$, as well as the functions $a_4(0, z)$, $f_4(0, z)$ and $\alpha(z)$ that determine the stress-energy tensor at initial time $t = 0$. The initial conditions for the electromagnetic field strength can be parametrised by $F_{ry}(r, 0, z) = \partial_r A_y$ and the normalisable mode of $A_t$, which we call $a_{t,2}(t, z)$. The initial conditions $B(r, 0, z)$ and $F_{ry}(r, 0, z)$ can be used to start with a far-from-equilibrium state, which then relaxes in a time of order $1/T$, where $T$ is the local temperature at the moment at which hydrodynamics becomes a good description (hydrody-namisation) [40]. In this work, however, we are interested in much longer time scales and we therefore set the initial values of these two functions to zero, i.e., their values in thermal equilibrium. The initial conditions for the cold and hot bath are then solely determined by the corresponding energy and charge densities, for which we choose

$$\mathcal{E}(z) = \mathcal{E}_C + (\mathcal{E}_H - \mathcal{E}_C)\,\theta\big(z - \tfrac{1}{4}z_{\max}\big)\,\theta\big(\tfrac{3}{4}z_{\max} - z\big)\,,$$
(49a)
$$\rho(z) = \rho_C + (\rho_H - \rho_C)\,\theta\big(z - \tfrac{1}{4}z_{\max}\big)\,\theta\big(\tfrac{3}{4}z_{\max} - z\big)\,,$$
(49b)

where we define $\theta(x) = \frac{1}{2}\big(1 + \tanh\frac{3}{2}x\big)$ to be a smooth step function and $z_{\max}$ denotes the size of the computational domain in $z$-direction. Since we neglect the back-reaction of the gauge field to the geometry our results are conformally invariant and the ratio of the energy densities of the hot and cold baths simply equals the fourth root of the ratio of the corresponding temperatures $\frac{\mathcal{E}_H}{\mathcal{E}_C} = \sqrt[4]{\frac{T_H}{T_C}}$. To make the scale invariance of our results manifest we multiply axis labels and legends by appropriate powers of $m = \pi T_C$, with $T_C = (\frac{4}{3}\mathcal{E}_C)^{1/4}/\pi$ being the temperature of the cold bath.

We close this section with some comments on the numerical scheme we use to solve the dual gravity problem. We impose periodic boundary conditions at $z = 0 = z_{\max}$. In the longitudinal direction we use a Fourier decomposition with 1500 grid points, whereas in the holographic direction we use a pseudo-spectral representation with 28 grid points. Our longest simulations use $z_{\max} = 80\pi$ and run from $t = 0$ till $t = 80$ with a time step of $\delta t = 0.0012$. In all our plots we shift one of the hot/cold transitions to the origin and make sure to only show times where the periodic boundary conditions do not yet affect the results. At every time step we apply low-pass filters to the time derivatives, whereby for the holographic direction we interpolate on a grid with 2/3 of the original grid points and subsequently interpolate back to main grid (see [36]). For the longitudinal direction we keep the lowest 30% of the Fourier modes used. Using *Mathematica 11* with the scheme presented in [38] this gives a runtime on a standard laptop of about one week for each of the runs presented.

---

it explicit that we work in a 3+1D boundary field theory. Also note that in Section 2 $\mathcal{E}$ referred to the energy density in the local restframe, whereas to be consistent with previous literature we here use $\mathcal{E}$ for the energy density in the lab frame. For the hot and cold bath the lab frame is the local restframe, but for the steady state and rarefaction waves it is necessary to compare to $T^{tt}$ in (1).

# 4 Holographic entanglement entropy

We consider entanglement entropy as a measure for the entanglement of states associated to different spatial subregions $\mathcal{R}$ in quantum field theory [41],

$$S_{\mathcal{R}} = -\text{Tr}_{\mathcal{R}} \hat{\rho}_{\mathcal{R}} \log \hat{\rho}_{\mathcal{R}}, \tag{50}$$

where $\hat{\rho}_{\mathcal{R}} = \text{Tr}_{\bar{\mathcal{R}}} \hat{\rho}$ denotes the reduced density matrix obtained by performing on the full density matrix $\hat{\rho}$ a partial trace over the degrees of freedom outside $\mathcal{R}$. For simplicity we will assume spatial subregions that are adapted to the symmetries of the Riemann problem. This means we choose for $\mathcal{R}$ at every constant time-slice ($t = t_0$) spatial stripes of finite width $\ell$ in $z$-direction and assume very large extend $\ell_\perp \gg \ell$ in the two other spatial directions $x_\perp^1$ and $x_\perp^2$,

$$\mathcal{R}_\pm = \{t = t_0, \ -\ell/2 \le z \mp \Delta z \le \ell/2, \ |x_\perp^1| = |x_\perp^2| \le \ell_\perp \}. \tag{51}$$

In practice we assume $\ell_\perp \to \infty$ and define two different regions $\mathcal{R}_\pm$ centered at a distance $\pm \Delta z$ to the left and to the right of the initial location $z = 0$ of the interface in the Riemann problem. In Fig. 4 we show a typical arrangement of entangling regions that we use in our numeric simulations. The two entangling regions with $\ell = 1$ are, shown in blue and red, are centered at $\Delta z = \pm 4$. We also show a typical initial (solid black) and late time (dashed black) profile of the energy density. Our motivation for this specific placement of the two

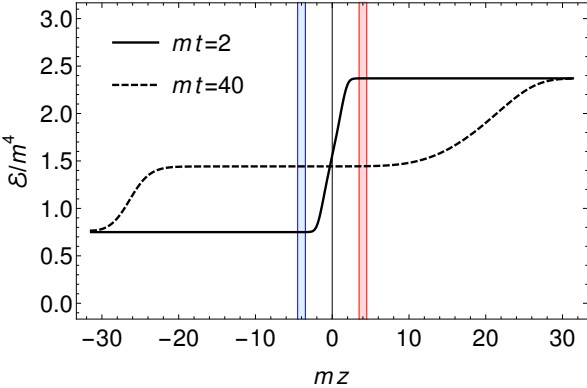

Figure 4: Entangling regions: blue and red stripes are typical arrangements of entangling regions of size $\ell = 1$ that initially reside entirely within the cold and hot bath, respectively. Black solid and black dashed lines are the spatial distributions of the energy density at early ($t = 2$) and late ($t = 40$) time in units of $m = \pi T_C$.

regions is that both regions reside initially entirely within either the cold or the hot bath, whereas at late time they both reside entirely within the NESS region. This will allow us to independently monitor the propagation of entanglement by the shock wave in the left region and the propagation of entanglement by the rarefaction wave in the right region and compare the results.

In the limit $\ell \to \infty$ the entangling region (51) covers an entire spacelike slice of Minkowski space and for thermal equilibrium states (50) equals the von Neumann entropy of the full density matrix $\hat{\rho}_{\mathcal{R}} = \hat{\rho}$, i.e., the thermodynamic entropy of a quantum state in thermal equilibrium. For the NESS system the situation is more subtle since at finite $t$ the NESS is still of finite size. The expectation, however, is that after taking $t \to \infty$ first it is possible to

take the large size limit $\ell \to \infty$ with the region falling entirely in the NESS regime. In that case it should be possible to identify the entanglement entropy with the thermal entropy of the boosted thermal state.

Explicit solutions for entanglement entropy are only available in exceptional cases such as free QFTs [42] or for 1+1 dimenstional CFTs in time-independent [43] and time dependent settings [44]. The holographic duality replaces the field theory computation of entanglement entropy by a much simpler extremisation problem for the area $\mathcal{A}_\mathcal{R}$ of a codimension two surface in the bulk [18],

$$S_\mathcal{R} = \frac{\mathcal{A}_\mathcal{R}}{4G_N} \,. \tag{52}$$

We emphasise that this was originally proposed in a static setting [17], where the extremisation reduces to a minimal surface problem, but was later extended to the time-dependent setting we use here. The relevant surface shares its boundary with the entangling region $\mathcal{R}$ in the field theory and extremises the area functional in the bulk theory

$$\mathcal{A}_\mathcal{R}[X] = \int \mathrm{d}^3\sigma \sqrt{\mathrm{Det}\left(\partial_a X^M \partial_b X^N g_{MN}\right)}, \quad \text{s.t.} \quad \partial X = \partial \mathcal{R} \,. \tag{53}$$

In general the surface embedding $X^M = X^M(\sigma^a)$ is parametrised by three intrinsic coordinates $\sigma^a$. In our context, it is convenient to switch from (44a) to the inverse radial coordinate $u = 1/r$ for which the boundary is located at $u = 0$,

$$ds^2 = g_{MN}\mathrm{d}x^M\mathrm{d}x^N = -C\mathrm{d}t^2 - \frac{2\mathrm{d}u\mathrm{d}t}{u^2} + 2G\mathrm{d}z\mathrm{d}t + S^2\left(e^B\mathrm{d}\mathbf{x}_\perp^2 + e^{-2B}\mathrm{d}z^2\right) \,, \tag{54}$$

with $\{C, G, S, B\}$ depend on $\{u, t, z\}$. The entangling regions (51) do not break translation symmetry in $x_\perp^{1,2}$-directions of the line element (54), hence also not of the Riemann problem in the boundary theory. Since we neglect the backreaction of the gauge field to the geometry, it does not enter in the calculation of the entanglement entropy. Analogously to [45] we can parametrise the bulk surface as follows,

$$X^M(\sigma, x_\perp^1, x_\perp^2) = \{X^\alpha(\sigma), x_\perp^1, x_\perp^2\}, \quad X^\alpha(\sigma) = \{U(\sigma), T(\sigma), Z(\sigma)\} \,. \tag{55}$$

This choice simplifies the area functional considerably, because the integration over the perpendicular directions $x_\perp^{1,2}$ can be performed explicitly and gives an overall factor

$$\iint\limits_{-\ell_\perp/2}^{\ell_\perp/2} \mathrm{d}\vec{x}_\perp = \ell_\perp^2 \,. \tag{56}$$

The remaining expression takes the form of a geodesic action,

$$\mathcal{A}_\mathcal{R}[X] = \ell_\perp^2 \int \mathrm{d}\sigma \sqrt{\bar{g}_{\alpha\beta}(U(\sigma), T(\sigma), Z(\sigma))\frac{\mathrm{d}X^\alpha}{\mathrm{d}\sigma}\frac{\mathrm{d}X^\beta}{\mathrm{d}\sigma}} \quad \text{s.t.} \quad X^\alpha(0) = \{0, t_0, \pm\ell/2\} \,, \tag{57}$$

where the metric $\bar{g}_{\alpha\beta}$ is related by a conformal factor to a three dimensional subspace ($\alpha, \beta = \{u, t, z\}$) of the bulk metric (44a)

$$\bar{ds}^2 = \bar{g}_{\alpha\beta}\mathrm{d}x^\alpha\mathrm{d}x^\beta = S(u, t, z)^4 e^{2B(u,t,z)} g_{\alpha\beta}\mathrm{d}x^\alpha\mathrm{d}x^\beta \,. \tag{58}$$

The equations of motion that follow from $\delta\mathcal{A}_\mathcal{R} = 0$ are given by the geodesic equation

$$\frac{\mathrm{d}^2 X^\alpha}{\mathrm{d}r^2} + \Gamma^\alpha_{\beta\gamma}\frac{\mathrm{d}X^\beta}{\mathrm{d}r}\frac{\mathrm{d}X^\gamma}{\mathrm{d}r} = J\frac{\mathrm{d}X^\alpha}{\mathrm{d}r}\,, \tag{59}$$

where $\Gamma^\alpha_{\beta\gamma}$ is the Levi-Cività connection associated to $\bar{g}_{\alpha\beta}$ and is meant to be evaluated at the location of the surface $X^\alpha(\sigma)$; the viscous friction term on the right hand side includes the Jacobian $J = \frac{\mathrm{d}^2\tau(\sigma)}{\mathrm{d}\sigma^2}/\frac{\mathrm{d}\tau(\sigma)}{\mathrm{d}\sigma}$ that originates from transforming from the affine parameter $\tau$ defined by $\frac{\mathrm{d}X^\alpha(\tau)}{\mathrm{d}\tau}\frac{\mathrm{d}X^\beta(\tau)}{\mathrm{d}\tau}\bar{g}_{\alpha\beta} = 1$ to the non-affine parameter $\sigma$. For numerical convenience, we choose a parametrisation that leads to the following Jacobian (for details see [46])

$$J(\sigma) = \frac{-51\sigma + 145\sigma^3 - 205\sigma^5 + 159\sigma^7 - 65\sigma^9 + 11\sigma^{11}}{(2-\sigma^2)(1-\sigma^2)(3-3\sigma^2+\sigma^4)(1-\sigma^2+\sigma^4)}\,. \tag{60}$$

The area functional (53) for the stripe region (51) suffers from two kinds of infinities. The first one is due to the infinite overall factor $\ell_\perp^2$ due to the infinitely long sides of the stripe in transverse direction. Since this factor contains no dynamical information we tame this infinity by considering in practice the entanglement entropy per transverse area $S_\mathcal{R}/\ell_\perp^2$. The second one is less trivial and due to the fact that extremal surfaces in the HRT-prescription of the holographic entanglement entropy (HEE) extend all the way to the asymptotic boundary, which has infinite distance from any point in the interior. To regularise the entanglement entropy we subtract the vacuum value, i.e., the area of surfaces in Poincaré patch $\mathrm{AdS}_{d+1}$ with appropriate conformal pre-factor,

$$\bar{d}s_0^2 = \bar{g}^{(0)}_{\alpha\beta}\mathrm{d}x^\alpha\mathrm{d}x^\beta = \frac{1}{u^{2(d-2)}}\left(\mathrm{d}t^2 - 2\mathrm{d}t\mathrm{d}u + \mathrm{d}z^2\right)\,. \tag{61}$$

The solution for the extremal surface embedding of stripe regions can be expressed in closed form

$$U_0(\sigma) = u_*(1-\sigma^2)\,, \tag{62a}$$

$$Z_0(\sigma) = \mathrm{sgn}(\sigma)\left(-\frac{\ell}{2} + \frac{U_0(\sigma)^4}{du_*^{d-1}}{}_2F_2\left[\tfrac{1}{2}, \tfrac{d}{2(d-1)}, \tfrac{3d-8}{2d-6}; \left(\frac{U(\sigma)}{u_*}\right)^{2(d-1)}\right]\right)\,, \tag{62b}$$

$$T_0(\sigma) = t_0 - U_0(\sigma)\,, \tag{62c}$$

where the $u_* = \frac{2\ell}{\sqrt{\pi}}\Gamma\left(\frac{1}{d(d-1)}\right)/\Gamma\left(\frac{d}{2(d-1)}\right)$ is the location of the turning point of the surface in radial direction. The corresponding cut-off regularised surface area is given by

$$\mathcal{A}_0^{\mathrm{cut}} = \ell_\perp^{d-2}\int_{\sigma_-}^{\sigma_+}\mathrm{d}\sigma\frac{1}{U_0^{d-1}}\sqrt{-\dot{T}_0^2 - 2\dot{U}_0\dot{T}_0 + \dot{Z}_0^2}\,, \tag{63}$$

where the cutoff at fixed radial location $u = u_{\mathrm{cut}}$ is realized by the following bounds on the non-affine parameter

$$\sigma_\pm = \pm\sqrt{1 - \frac{u_{\mathrm{cut}}}{u_*}}\,. \tag{64}$$

Together with the cut-off regularised expression for the gravity dual of the Riemann problem

$$\mathcal{A}^{\mathrm{cut}} = \ell_\perp^2\int_{\sigma_-}^{\sigma_+}\mathrm{d}\sigma S^2 e^B\sqrt{-A\dot{T}^2 - \frac{2}{U^2}\dot{U}\dot{T} + 2F\dot{T}\dot{Z} + S^2 e^{-2B}\dot{Z}^2} \tag{65}$$

we can express the finite vacuum subtracted entanglement entropy per transverse area as

$$S_{\text{ren}} = \frac{\mathcal{A}^{\text{cut}} - \mathcal{A}_0^{\text{cut}}}{4 G_{\text{N}} \ell_\perp^2} \,. \tag{66}$$

In practice, we solve (59) using a relaxation algorithm [27, 46] using a cut-off $u_{\text{cut}} = 0.075$ and verified this value is small enough to not affect the results presented.

## 5 Results

### 5.1 Energy and charge density

In this section we present our results for the evolution of energy and charge density obtained from the holographic calculation. The global features of the evolution are similar to those of the analytic shock+rarefaction wave solution of the Riemann problem obtained in Sec. 2.2. In Fig. 5 we plot the time evolution of the energy and charge density for an initial cold/hot ratio of $\chi = \sqrt{\mathcal{E}_C/\mathcal{E}_H} = \sqrt{n_C/n_H} = 9/16$ for both the energy and charge density. In the plot

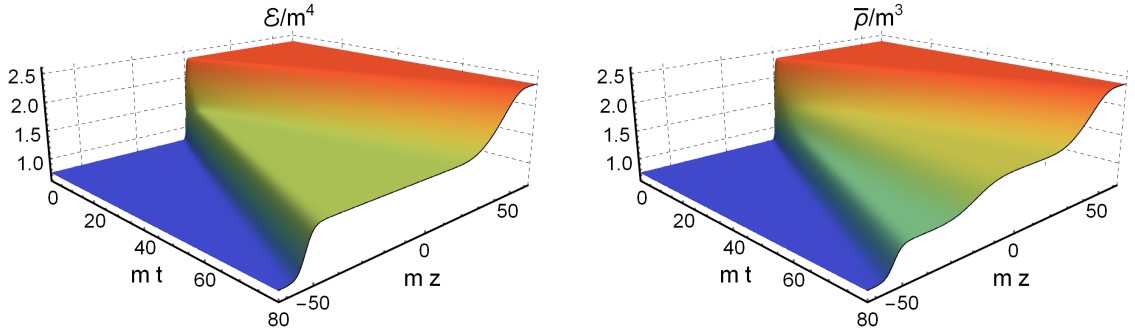

Figure 5: Time evolution of the energy density (left) and charge density (right) for $\chi = 9/16$ and $n_C/n_H = \chi^2$. A steady state region forms between the shock (moving towards the cold bath) and the rarefaction wave (moving towards the hot bath). Two regions with constant charge density can be identified within the steady state region.

for the energy density (left) we clearly see a NESS region emerging briefly after $t = 0$ between two wave fronts that propagate from $z = 0$ towards $z = \pm\infty$. In the plot on the right we show the evolution of the charge density. Two regions with constant but different charge densities emerge inside the NESS region, which indicates the formation of a contact discontinuity.

It is interesting to compare the result of the holographic simulation to the solution of the corresponding Riemann problem in ideal hydrodynamics. For this we show in Fig. 6 profiles of the energy density and charge density of the holographic result at various times together with the (unphysical) shock+shock and (physical) shock+rarefaction solutions presented in Sec. 2. In addition we include a numerical solution obtained from an ideal hydrodynamic simulation with smooth initial conditions for the energy density

$$\mathcal{E}(z) = \frac{\mathcal{E}_C + \mathcal{E}_H}{2} + \frac{\mathcal{E}_H - \mathcal{E}_C}{2} \tanh\left( \frac{z_{\max}}{20} \sin \frac{2\pi z}{z_{\max}} \right) \,. \tag{67}$$

The factor 20 in the denominator of (67) is a convenient numerical choice for realizing the initial conditions. The shapes of the shock and the rarefaction wave in the ideal hydro

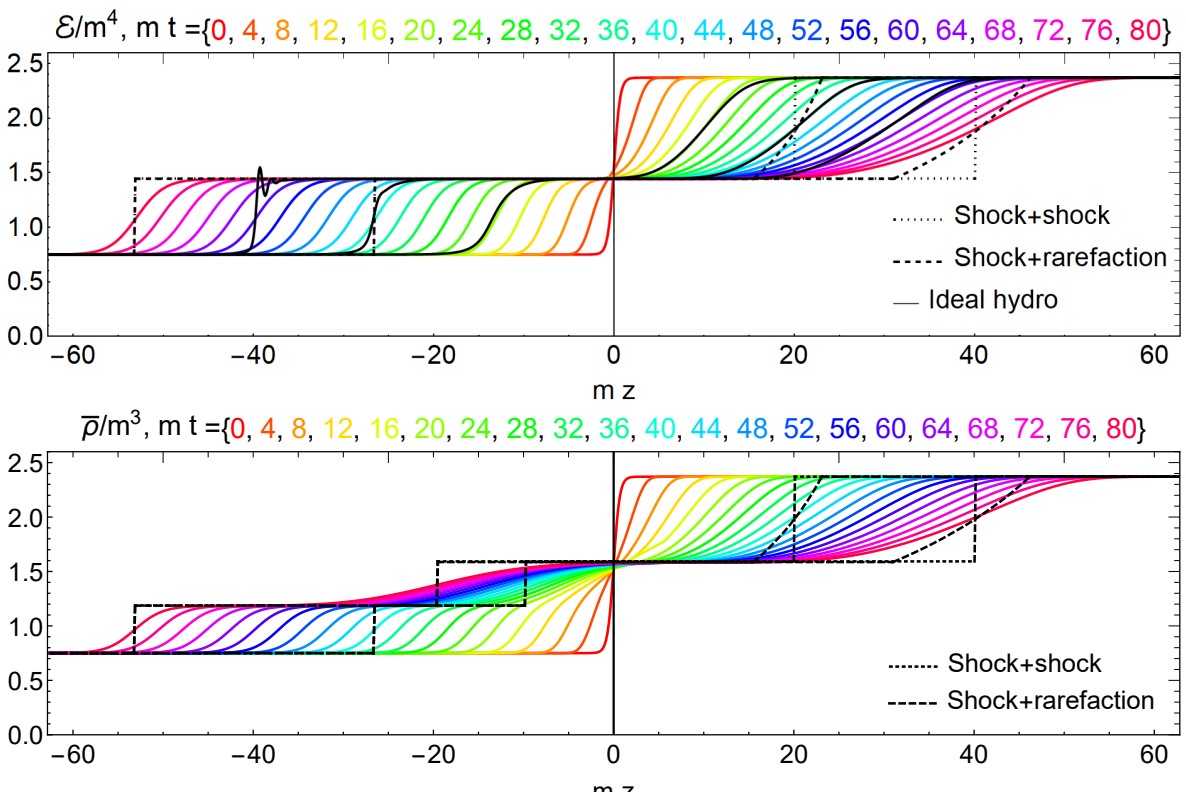

Figure 6: Evolution of the energy density (top) and charge density (bottom). Coloured lines are snapshots at various different times of the holographic simulation. For comparison we show in the upper panel results of the analytic shock+shock (black dotted) and shock+rarefaction (black dashed) solution as well as a numeric solution of the ideal hydrodynamics equations with smooth initial data (black solid).

simulation depend on the initial details of the transition region. However, we have checked that the energy density in the steady state region is insensitive to this choice. In Appendix B we verify that also the properties of the NESS region in the holographic system do not depend on how the initial conditions are set up.

Although the initial conditions of the hydrodynamic simulation are perfectly smooth, the wave traveling towards the cold side steepens significantly as time progresses until the applicability of ideal hydrodynamics and eventually also the numerical evolution breaks down. The formation of shocks from smooth initial data is a well known phenomenon in non-viscous hydrodynamics and faithful simulations require shock-capturing methods which we did not attempt to implement. The wave that moves towards the hot side on the other hand remains smooth and becomes wider with time. Coloured lines present snapshots of the holographic result, which at early times resemble the hydrodynamic solution, but at late times become closer to the shock+rarefaction solution. The energy density in the NESS steady state region agrees accurately with the result from the ideal hydrodynamic calculation, but differs slightly from the analytic shock+shock (dotted) solution. To be precise, using (27) we find $\mathcal{E}_{\mathcal{S}} = 625/432 \approx 1.44676$, $\mathcal{E}_{\mathcal{S}} \approx 1.4450$ and $\mathcal{E}_{\mathcal{S}} \approx 1.446$ for the shock+shock, shock+rarefaction and ideal hydrodynamics case respectively and $\mathcal{E}_{\mathcal{S}} \approx 1.4435$ in the holographic simulation. By varying the $z$-location of the probe point we estimate the numerical accuracy of the holographic result to be 0.001, which means that within numerical accuracy the holographic result agrees with the shock+rarefaction solution as presented in Section 2.

In the bottom panel of Fig. 6, we plot profiles of the charge density in the holographic result (solid coloured lines) at various times together with two dashed lines for times $t = 40$ and 80, which have $\bar{\rho}_1 = 19/16 = 1.1875$ and $\bar{\rho}_2 = 43/27 \approx 1.5926$ obtained from the analytic double shock solution (11) as well as two dotted lines at $\bar{\rho}_1 \approx 1.1866$ and $\bar{\rho}_2 \approx 1.5866$ of the shock+rarefaction solution (31). We evaluate the full holographic values of $\bar{\rho}_{1,2}$ at the point where the $z$-derivative of $\bar{\rho}$ is minimised, which at $t = 80$ happens at $z = -41.29$ and $z = 10.26$, where $\bar{\rho}_1 \approx 1.1845$ and $\bar{\rho}_1 \approx 1.5823$ respectively. In contrast to the analytic solution the contact discontinuity (see Sec. 2 and also [20]) manifests in the holographic model as a smooth crossover region that progressively broadens with time.

Fig. 7 shows the local fluid velocity, as determined by diagonalising the full stress-energy tensor (solid), as well as the velocity of the charge as defined by $v_{\text{charge}} = J_z/J_t$ (dashed). As expected from a solution that is locally in equilibrium the two velocities are virtually indistinguishable. In particular this implies that in the rest frame of the steady state the charge is almost at rest throughout the contact discontinuity. Nevertheless, due to diffusion the charge does smooth out, and a small velocity profile is visible (Fig. 7 bottom, see also Appendix B).

The time evolution of the charge and energy density and in particular the rarefaction wave can be better understood by showing the profiles as a function of the scaled coordinate $\xi = z/t$, as shown in Fig. 8. Since the width of the shock is approximately constant in time, in the scaled coordinates the shock indeed resembles more closely a true shock (i.e. a discontinuity), both for the energy and charge density. Also the rarefaction wave resembles the analytical rarefaction wave from Section 2 more closely at later times, but with the limited time span available it is not clear if it would converge to the analytic result in the late time limit. The charge density again clearly shows the two plateaus at late times, and also here it is clear that the diffusion at the contact discontinuity becomes a true discontinuity in the late time limit in these scaled coordinates (see Appendix B).

As explained in Sec. 2 the profile of the charge density depends on both the ratio of the

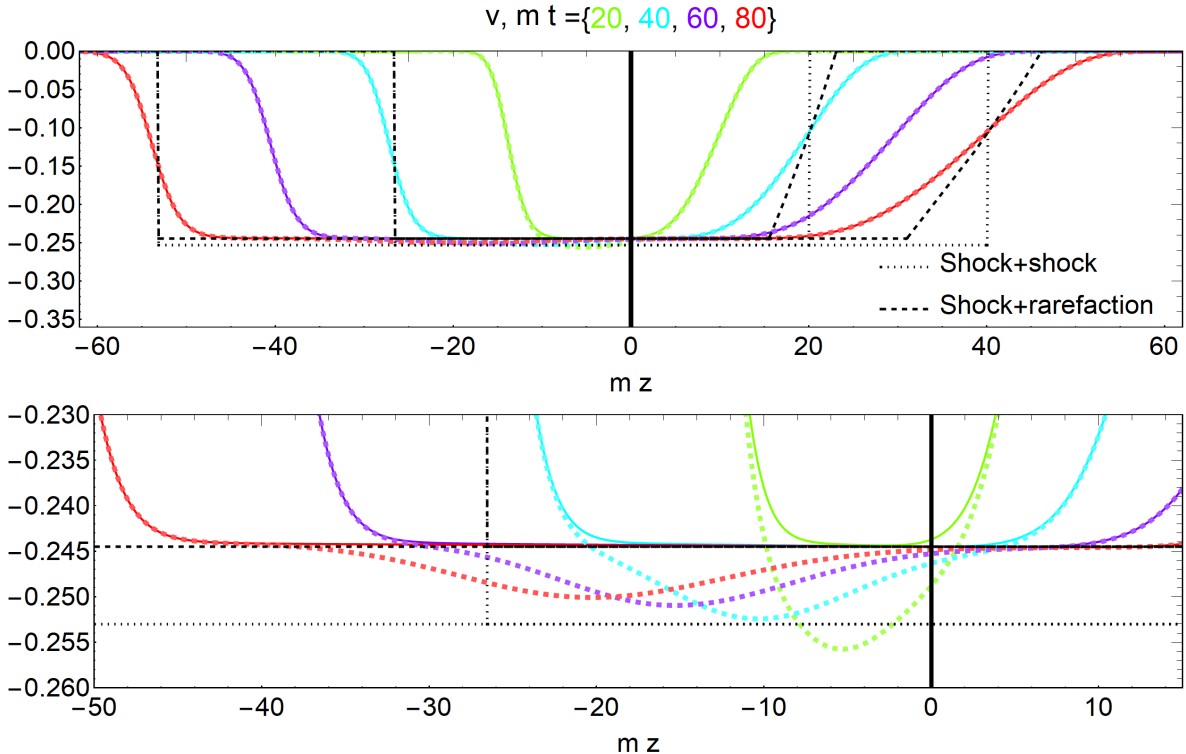

Figure 7: Snapshots of the local fluid velocity (solid) and charge velocity (dashed) at four different times for the same evolution as in Fig. 5. The bottom panel shows a magnification of the upper panel, where the velocity profile of the charge diffusion at the contact discontinuity of the two charge plateaus is clearly visible.

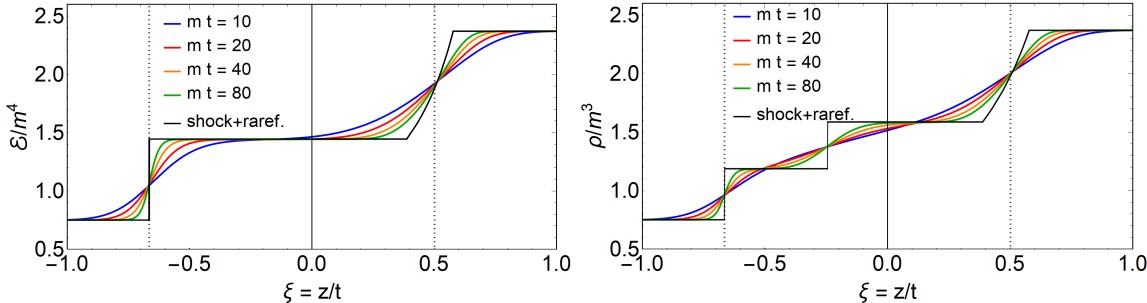

Figure 8: Time evolution of the energy density (left) and charge density (right) as a function of $\xi = z/t$. At the latest time we managed to obtain ($mt = 80$) the solution closely resembles the shock+rarefaction solution found in Section 2.

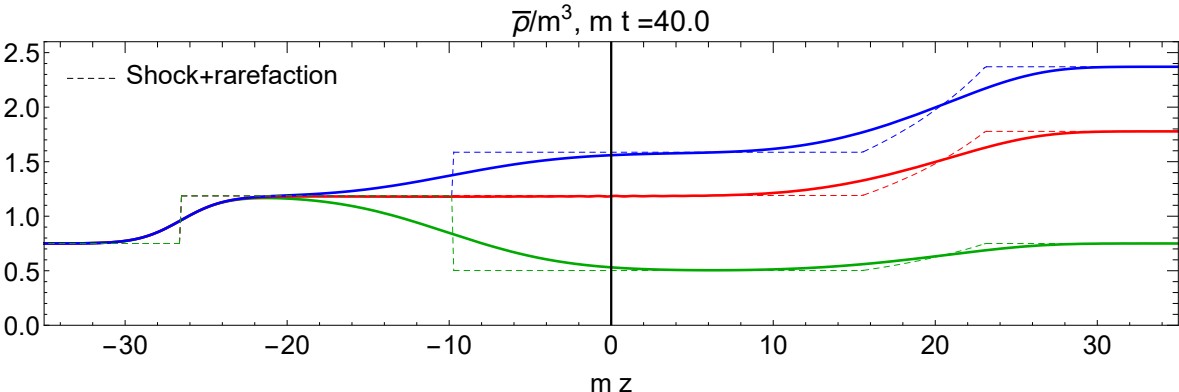

Figure 9: We show a snapshot of the charge density at $t = 40$ for a temperature ratio of $\chi = 16/9$ and three different ratios of the charge density, with $n_H/n_C$ being 256/81, 64/27 and 1 for the blue, red and green curves respectively. In general there are two regions of constant charged, determined solely by $\chi$ and $n_H/n_C$. These two regions almost coincide for the 64/27 ratio, but are different for the red and green evolutions. For comparison we again show the analytical shock+rarefaction solutions of Section 2 (dashed).

$\chi$ as well as $n_C/n_H$. In Fig. 9 we show charge densities at $t = 40$ for $n_C/n_H$ equal to 256/81 (blue, same as Fig. 5), 64/27 (red) and 1 (green). The ratio 64/27 equals $\chi^{-3/2}$, which implies that the third root of the charge density has the same ratio as the fourth root of the energy density, as suggested by dimensional analysis. Indeed this ratio approximately leads to a charge profile with only one constant charge region, as opposed to the other two solutions, where two separate charged regions are present. Even if the charge densities in the hot and cold bath are equal ($n_C = n_H$) there is still a non-trivial charge flow and density, as driven by the steady state of the energy density.

One crucial feature of the NESS is that in particular the shock is a far-from-equilibrium effect that cannot be described by hydrodynamics. To quantify this we show in Fig. 10 (left) the transverse pressure over the energy density in the local restframe (again determined by diagonalising the stress-energy tensor). In ideal hydrodynamics of a conformal theory this ratio $\mathcal{P}_T/\mathcal{E}_{\mathrm{loc}} = 1/3$, however in particular near the shock region significant deviations are visible (see also solid lines in Fig. 10 (right)). Using the hydrodynamic constituent equations and our determined local energy density, fluid velocity as well as the ratio $\eta/s = 1/4\pi$, we also compare the transverse pressure with the transverse pressure as determined from first order viscous hydrodynamics, as shown as dashed lines in Fig. 10 (right). Indeed the evolution of the rarefaction wave can be entirely described using viscous hydrodynamics, however for the shock there are significant differences.

Numerically it is more challenging to evolve profiles with $\chi < 9/16$. However, we were able to evolve some shorter runs stably, albeit on smaller grids and with a shorter evolution time. Results for the energy density and the hydrodynamic comparison at $t = 12$ are shown in Fig. 11 for values of $\chi$ of 9/16 (blue, as in Fig. 5), 4/9 (red) and 16/49 (green). Interestingly, also for these larger ratios the rarefaction wave is always well described by hydrodynamics, though with larger viscous corrections as is apparent from the comparison with ideal hydrodynamics. For the shock region deviations from viscous hydrodynamics become larger as one increases the ratio between the hot and cold baths.

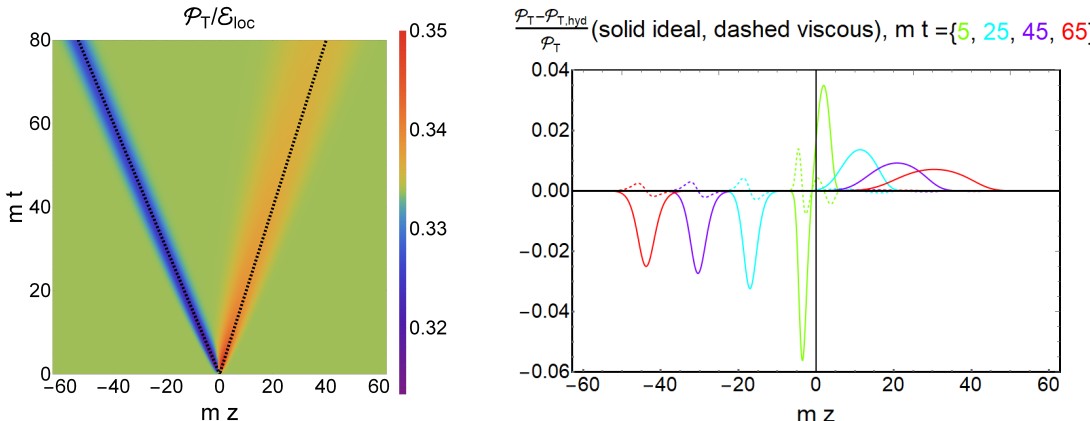

Figure 10: (left) For the evolution of Fig. 6 we show the ratio of the transverse pressure over the energy density in the local restframe together with the shock velocities (black, dashed) computed from Eqn. (11). Most of the evolution is close to thermal equilibrium ($\mathcal{P}_T = \mathcal{E}_{\mathrm{loc}}/3$, in green in the colour coding), but around the shocks there are significant deviations. (right) For several times we present the deviations with respect to ideal hydro (solid) and first order viscous hydrodynamics (dashed), normalised to the transverse pressure itself. The entire evolution can be described by viscous hydrodynamics with better than 1% accuracy.

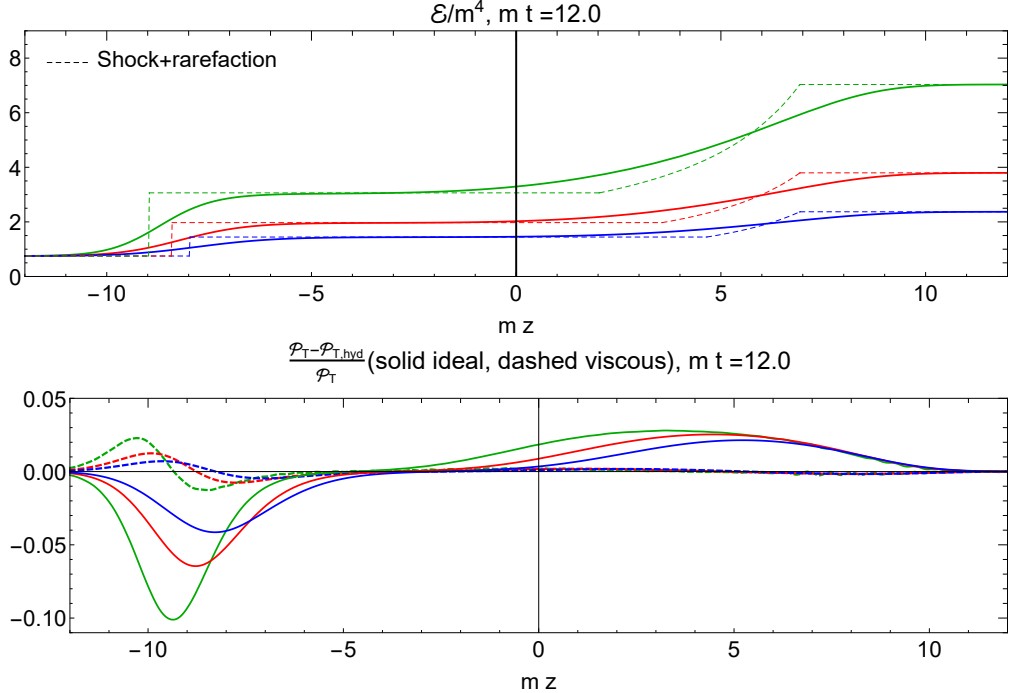

Figure 11: (top) Energy density for different heat bath temperature ratios together with the shock+rarefaction from Section 2 (dashed). (bottom) Comparison with ideal (solid) and viscous (dashed) hydrodynamics. For larger ratios ideal hydrodynamics becomes a worse description for the rarefaction wave, though viscous hydrodynamics is applicable. The shock region has increasingly large deviations from viscous hydrodynamics as the ratio is increased.

## 5.2 Shock evolution and entropy production

One of the main motivations in studying the NESS in holography is a complete description of its dynamics beyond the hydrodynamic limit, which is particularly relevant for the evolution of the shocks. References [15,20] showed that at intermediate times the shock widens diffusively as governed by viscous hydrodynamics, with characteristic width $w_{\text{shock}} \propto \sqrt{t}$. At some time, however, the entropy production within hydrodynamics is not large enough to be consistent with the total entropy production and the shock cannot continue to diffuse. The typical timescale for this transition was estimated to be equal to

$$t_{\text{diff}} \propto \frac{\eta}{sT\delta^2}, \tag{68}$$

where $\delta = T_H/T_C - 1$ is assumed to be small. After this time it is unknown if the shock settles down to a soliton-like object of constant width, or if it continues to widen at a smaller rate. In Fig. 12 (left) we show snapshots of the time derivative of the energy density for the evolution of the shocks in Fig. 5 (which has $\delta = 1/3$). In the right plot we show the full-width at half-maximum of the shocks shown in the left plot as a function of time. Indeed, at early times the width grows diffusively (red dashed, with a small off-set that is subleading, but can also be partly explained by the initial profile). Around $t \approx 20$ the width starts growing more slowly, in qualitative agreement with the estimate provided in (68) (note that $T = m/\pi$). All our numerical data points indicate that this width keeps growing logarithmically in time, but we note that it is nevertheless also possible to fit a function of the form $C + ae^{-bt}$, which would settle down to the soliton-like object as conjectured in [20].

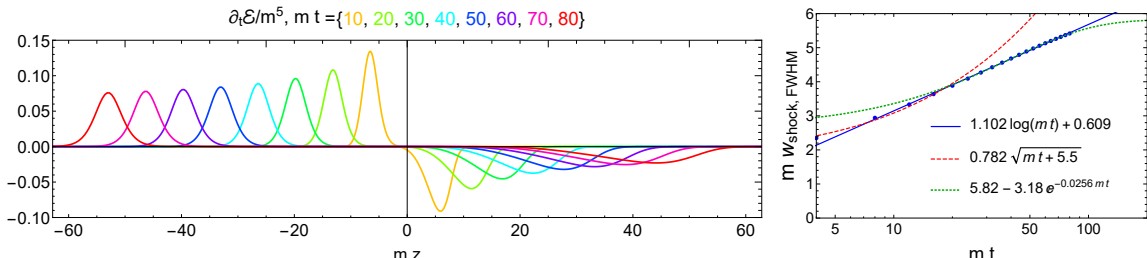

Figure 12: We show the change in shape of the outgoing waves from the time derivative of the energy density. Colours from yellow to red correspond to snapshots of the energy density profile from early to late times. Both, the left moving and the right moving waves disperse. On the right we show the time evolution of the full-width at half-maximum together with different numerical fits, corresponding to diffusive growth (red dashed), logarithmic growth (blue solid), or a possible exponential decay to a shock of constant width at late times (green dotted).

As discussed in section 2, solutions to the Riemann problem in ideal hydrodynamics are not unique and physically sensible solutions need to be selected by imposing additional constraints, such as the entropy condition (13). Solutions to the holographic Riemann problem are unique and conditions such as (13) are encoded in the equations of motion of the dual gravity problem [47]. In Fig. 13 we show snapshots of the divergence of the entropy current as a function of time, where $s^\mu = \frac{4\mathcal{E}_{\text{loc}}}{3T}u^\mu$, with $u^\mu$ the local fluid velocity and $\mathcal{E}_{\text{loc}}$ the energy density in the local restframe. The entropy production is negligible in the NESS region, but a significant amount of entropy is produced by the outgoing waves. Fig. 13 (right) shows the integral of the divergence of the entropy current over the waves that travel towards the cold (blue dots) and the hot side (red dots). The entropy produced by the wave moving towards the cold

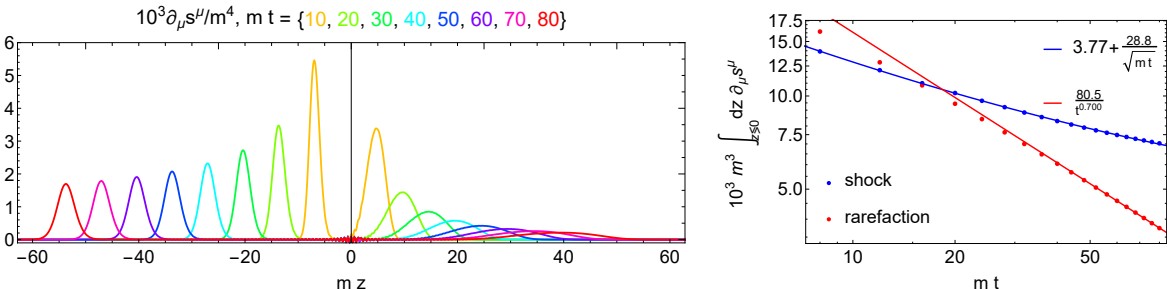

Figure 13: We show the divergence of the entropy current for several times (left) as well as the integral over the left (shock, blue) and right (rarefaction, red) regions (right). The entropy production in the shock settles down to a constant value, whereas for the rarefaction wave the entropy production decays to zero in a power-law fashion.

bath slowly decays to a constant value, similar to a shock wave, and can be compared to the analytic solution (14), which for $\chi = 9/16$ gives

$$\int_{z<0} dz\, \partial_\mu s^\mu = \frac{\pi}{\sqrt{3}m^3} \left( \chi^{-1/4} - \sqrt{\frac{\chi(3+\chi)}{1+3\chi}} \right) \approx 0.00610. \tag{69}$$

At the end of the simulation we find an entropy production of about 0.007, which is still higher than the value obtained from the shock+shock solution of the Riemann problem cited above, but the extrapolation shown in Fig. 13 predicts for $t \to \infty$ a significantly smaller final value of about 0.0037. On the other hand, the entropy production of the wave moving towards the hot side (red dots in Fig. 13 (right)) decays to zero with a power law indicating that this wave indeed becomes at late time a rarefaction wave which has zero entropy production per definition.

Lastly, we can compare the total entropy production directly with the holographic dual by evaluating the area density of the apparent horizon $\mathcal{A}$ as shown in Fig. 14 (left), whereby we use that $\mathcal{S}_{\mathrm{AH}} = \mathcal{A}/4G_N$, and in analogy with (47) we show $s_{AH} \equiv \mathcal{S}/4\pi G_N$. In Fig. 14 (right) we show the spatial integral of the time derivative of the apparent horizon density and it can be seen that the time evolution matches well with the sum of the entropy production of the shock and rarefaction waves from the hydrodynamic results shown in Fig. 13.

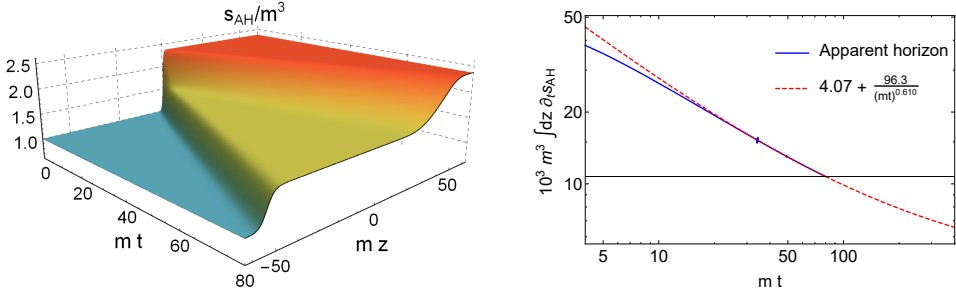

Figure 14: We show the entropy density as determined from the area of the apparent horizon (left) as well as time derivative of the spatial integral (right). The entropy production is to a good approximation given by the sum of the entropy production of the shock and rarefaction waves as derived in Fig. 13, and also the late time value (4.07, from the red dashed fit on the right) matches well with the hydrodynamic result shown in Fig. 13 (3.77).

## 5.3 Extremal surfaces and entanglement entropy

Let us now discuss our numerical results for the entanglement entropy in the NESS system. It is useful to first analyse some features of extremal surfaces from which we compute the entanglement entropy. Fig. 15 shows a typical family of such surfaces together with the radial position of the apparent horizon in a gauge where $\alpha = 0$. In the top (bottom) row, we display the results for entangling regions of width $\ell = 2$ ($\ell = 1.5$) centered at $z = -4$ ($z = +4$) corresponding to the blue (red) region in Fig. 4. Surfaces with small $\ell$ (not shown here) reside mostly in the asymptotic AdS part of the geometry which explains the universal (state-independent) UV scaling of entanglement entropy. Surfaces with large $\ell$ reach deep into the bulk and are therefore sensitive to the geometry close to the horizon and lead to state-dependent contributions in the IR scaling of the entanglement entropy (see similar discussion in [48]).

In static and boosted AdS black brane geometries, extremal surfaces that are connected to the boundary cannot enter the region beyond the horizon [49]. However, in time-dependent geometries such as the one considered here, the situation is different and there are examples known, where extremal surfaces cross the apparent horizon and therefore also the event horizon in regions where the spacetime changes rapidly in time [50]. The holographic NESS system in our work is similar to a system of colliding shock waves [45], where geodesics dual to two-point functions could cross the horizon, but no such extremal surfaces were found. While we have not attempted to construct examples for such geodesics, we expect the situation to be similar in the holographic dual of the NESS system.

A further effect is the warping of surfaces close to the apparent horizon in the boosted part of the geometry, i.e. in the part that corresponds to the NESS in the boundary theory. This effect is clearly visible in our example with $\ell = 2$ (less pronounced for $\ell = 1.5$) in the plots in the middle of Fig. 15, in which we zoom into the region close to the horizon where the geometry transitions from the static to the boosted black brane geometry.

In addition, we show in the right panel of Fig. 15 cross-sections of surfaces in the $u$-$z$ plane at early ($t = 2$) and late time ($t = 40$), i.e., surfaces that reside entirely in the static and in the boosted black brane geometry, respectively. At early times, when the surfaces reside entirely in the static part of the geometry, the embedding function is symmetric with respect to the center of the entangling region (located at $z = \pm 4$). At late times this symmetry is clearly broken by velocity of the steady state.

In Fig. 16 we show the renormalised entanglement entropy as a function of time at these locations for several different lengths of the interval, ranging from $\ell = 0.6$ till $\ell = 2.0$, together with the energy density as also shown in Fig. 6. The renormalised entropy is computed as a difference to its vacuum value, for which we use a cut-off in the holographic coordinate at $u_{\mathrm{cut}} = 0.075$, in a gauge where $\alpha = 0$ (see also Sec. 3 and [45]). Since the entanglement entropy of the infinite strip region has both UV and IR divergences, we choose to show a linear transformation such that its renormalised version $S_{\mathrm{ren}}$ agrees with the energy density in both the black brane and the steady state regime: $aS_{\mathrm{ren}} + b = \mathcal{E}$, which in particular means the curves are insensitive to our choice of regularisation. The plots in Fig. 16 hence compare the shape of the entanglement entropy with the shape of the energy density both during the passing of the shock and rarefaction waves. From the insets it is clear that the entanglement entropy is slightly delayed as compared to the energy density, in particular for shorter intervals. This delay is expected, as the surfaces probe into the past geometry (Fig. 15), even though for larger intervals we note that the entangling geometry starts to feel

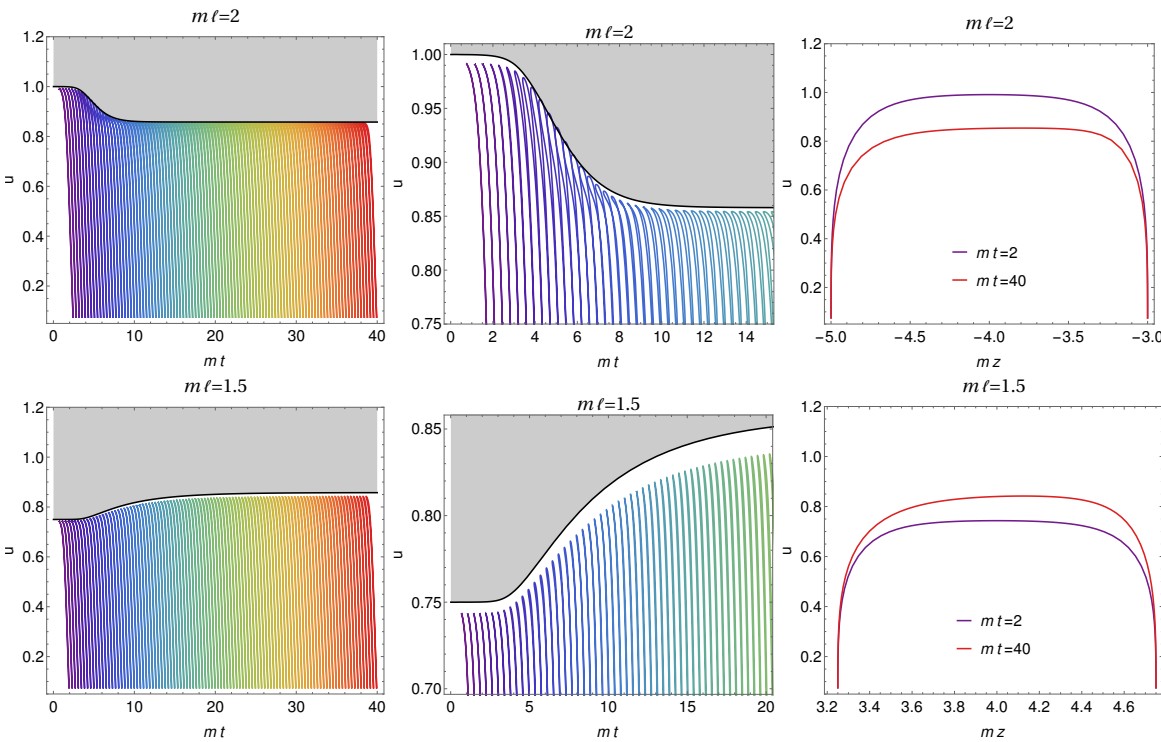

Figure 15: Cross-sections of extremal surfaces for entangling regions of size $\ell = 2$ centered at $z = -4$ (top, cold region) and of size $\ell = 1.5$ centered at $z = 4$ (bottom, hot region) in the geometry with $\chi = \sqrt{\mathcal{E}_C/\mathcal{E}_H} = 9/16$. The plots on the left show surfaces at various different boundary times and their position relative to the apparent horizon whose radial position is shown in black and regions beyond the horizon are shown in gray. In the middle, we zoom into the transition region close to the horizon between static and boosted black brane. On the right, we show cross-sections of surfaces in the $u$-$z$ plane at early ($t = 2$) and late time ($t = 40$).

the wave earlier, since the region is bounded by $z \pm \ell/2$.

Fig. 17 shows the renormalised entanglement entropy as a function of time for intervals of $\ell = 1$ located at eleven different locations ranging from $z = -4$ till $z = -14$ (left figure, shock region) and from $z = 4$ till $z = 14$ (right figure, rarefaction region). Here the same linear transformation is used as in Fig. 16 for $\ell = 1$. As time progresses the shock (left) or rarefaction (right) wave passes through the interval, after which the interval settles down to the steady state regime. The values of the entanglement entropy in the cold, steady state and hot bath regions are given by 0.466, 0.835 and 1.351 respectively, as can also be analytically computed [51]. We again find a small delay of the entanglement entropy evolution, which is more pronounced for the rarefaction case.

The time evolution of the entanglement entropy received some recent attention as a probe of equilibration towards a thermal state. After perturbing or quenching a quantum state, the entanglement entropy will saturate to its final value in a time $t_S$ that is at least $t_S \geq R/v_B$ [52], with $R$ the radius of the largest sphere that can be inscribed in the entangling region and $v_B$ the butterfly velocity that characterises chaotic growth of quantum operators (for our case of a neutral holographic plasma $v_B = \sqrt{d/2(d-1)}$ in $d$ spacetime dimensions). The start of this equilibration process, also called entanglement tsunami [53], is characterised by the entanglement velocity, whereby $S_{\text{EE}}(t) = s_{\text{eq}} A v_{\text{E}} t$, with $s_{\text{eq}}$ the equilibrium entropy density,

$A$ the area of the boundary of the entangling region and where this equation defines the entanglement velocity $v_E$. For neutral holographic plasmas it was found that [53]

$$v_E = \sqrt{d}(d-2)^{1/2-1/d}/(2[d-1])^{1-1/d} \tag{70}$$

and it was shown that for any theory $v_E \leq v_B$ [52].

For the case of the steady state formation, a simple approximation of $v_E$ is possible when the entanglement equilibration is much faster than the timescale of the perturbation of the state [16]. For the shock/rarefaction regions considered here, this would be the case when the respective shock and rarefaction velocities are much slower than $v_E$. In that case the time evolution of the HEE can be approximated by the time evolution of the equilibrium entropy density, which in the local restframe is just proportional to $T^d$. In this approximation the analogy of the entanglement velocity is given by [16]

$$v_{\text{av,C/H}} = v_{C/H}\left(1 - \frac{T_{C/H}^d}{\cosh(\theta)T^d}\right), \tag{71}$$

where $\theta = \text{arctanh}(v_S)$ is the boost factor associated to the steady state region. As discussed for the case of two shock waves in [16], when $\chi \to 0$ this velocity violates the bound on $t_S$ mentioned above for $d > 2$ spacetime dimensions. For $\chi \to 0$ a full holographic calculation is therefore necessary, extending the results for intermediate values of $\chi$ shown here in Figs. 16 and 17. Our small time delay of the HEE compared to the energy density can indeed be interpreted as the need to study the full HEE instead of the equilibrium entropy density. Unfortunately in our setting it is numerically difficult to probe small enough $\chi$ and large enough entangling regions to truly investigate the butterfly bound $t_S \geq R/v_B$. We leave this to a future investigation. A promising approach for this is to use membrane theory [52, 54].

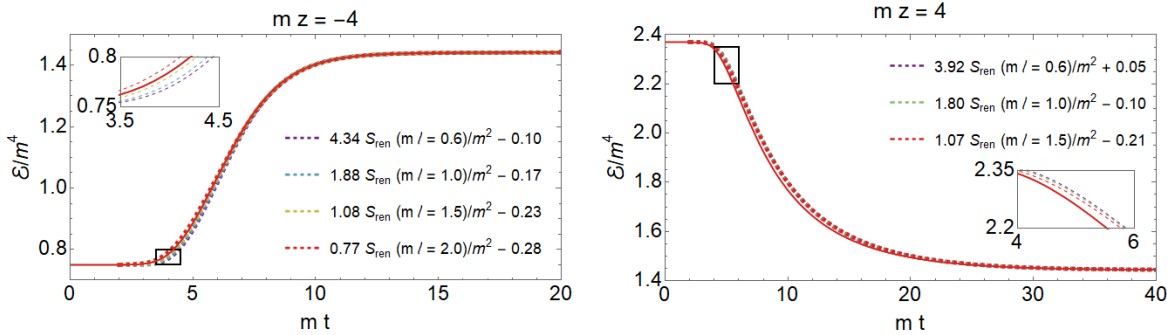

Figure 16: We show the energy density at $z = -4$ (shock regime, left) and $z = 4$ (rarefaction regime, right) together with the time evolution of the holographic entanglement entropy $S_{\text{EE}}$ for different lengths. Since the $S_{\text{EE}}$ is sensitive to UV regularisation and length dependence we apply a linear transformation such that at early time (hot/cold) and late time (steady state) the entanglement entropy agrees with the energy density. After this rescaling the curves agree almost exactly, although shorter lengths that are more sensitive to regions closer to the boundary have a small delay (see insets).

We end this section with an attempt to characterise the differences between shock and rarefaction waves in the NESS system with a new quantity that is inspired by the entanglement temperature.

In the limit where the entangling region is small compared to the length scale as given by the energy density (meaning $\ell^d \ll \mathcal{E}$) changes in entanglement entropy and the energy density

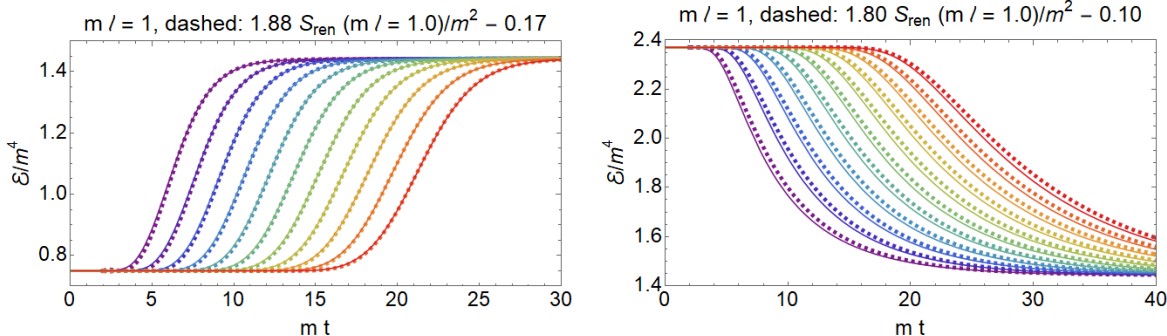

Figure 17: Similarly to Figure 16, for $\ell = 1$ we now show the energy density (solid) as well as the rescaled HEE (dashed, the rescaling parameters are the same as in Figure 16 for $\ell = 1$) for different positions of the entangling region, varying from $-4$ till $-14$ (left, shock region) and from $4$ till $14$ (right, rarefaction region). Note that the rescaling is the same for all curves, which reaffirms that the evolution of the HEE is almost entirely determined by the evolution of the energy density.

satisfy a universal relation that is analogous to the first law of thermodynamics ($dE = TdS$) and is therefore called first law of entanglement entropy [55]

$$\Delta E_{\mathcal{R}} = T_{\text{ent}}\Delta S_{\mathcal{R}}\,, \tag{72}$$

where $T_{\text{ent}}$ is the entanglement temperature, $\Delta S_{\mathcal{R}}$ is the variation of the entanglement entropy associated to the region $\mathcal{R}$ and $\Delta E_{\mathcal{R}}$ is variation of the integral of the energy density over $\mathcal{R}$. In this small size limit the entanglement temperature depends only on the theory and the shape of the chosen subregion, with

$$T_{\text{ent}} = c/\ell\,, \tag{73}$$

For stripe shaped subregions (51) and thermal states the constant $c$ can be expressed in closed form [55]

$$c = \frac{2(d^2 - 1)\Gamma\left(\frac{1}{2} + \frac{1}{d-1}\right)\Gamma\left(\frac{d}{2(d-1)}\right)^2}{\sqrt{\pi}\Gamma\left(\frac{1}{2(d-1)}\right)^2\Gamma\left(\frac{1}{d-1}\right)}\,, \tag{74}$$

while in the limit $\ell \to \infty$ the entanglement temperature becomes equal to the thermodynamic temperature $T_{\text{ent}} = T$.

Inspired by (72) we define the dynamic entanglement temperature as the ratio of the renormalised entanglement entropy and the total energy inside the entangling region

$$T_{\text{ent}}^{\text{dyn}} = \frac{\int_{\mathcal{R}} \mathrm{d}z \langle T^{tt}(t, z)\rangle}{S_{\text{ren}}(t)}\,. \tag{75}$$

The quantity (75) is well-defined for dynamic states and reduces for static states in the small $\ell$ limit to the entanglement temperature defined by (72). We compute $T_{\text{ent}}^{\text{dyn}}$ in the NESS with $\chi = \sqrt{\mathcal{E}_C/\mathcal{E}_H} = 9/16$ for entangling regions that are passed by shock and rarefaction waves. It turns out $T_{\text{ent}}^{\text{dyn}}$ behaves qualitatively different when either a shock or a rarefaction wave passes the region. The results for three different sizes ($\ell = 0.6, 1, 1.5$) of the entangling region are shown in Fig. 18. For these lengths (73) implies $T_{\text{ent}} = 0.703, 0.422, 0.281$ respectively,

and note that we have $T_C = 1/\pi \approx 0.318$, $T_S \approx 0.367$ and $T_H = 4/3\pi \approx 0.424$. From the figure it is clear that for $\ell = 0.6$ the dynamic entanglement temperature is closer to the small region limit (Eqn. (73)), whereas for $\ell = 1.5$ the result is closer to the large region limit (the physical temperature), and we see that the dynamic entanglement temperature is always higher than either of them. One curious feature happens when a rarefaction wave passes a

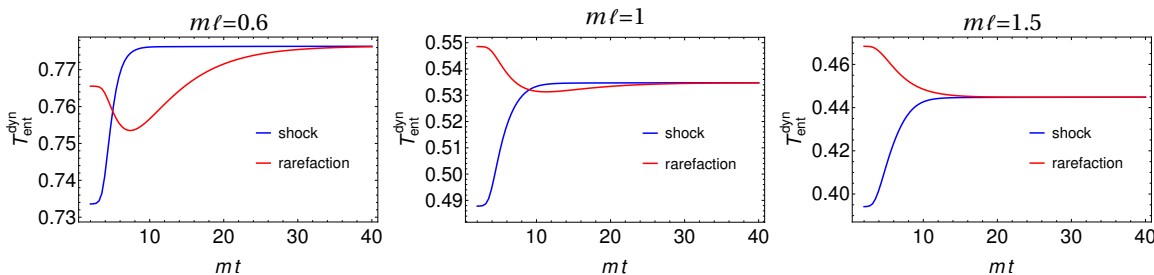

Figure 18: Dynamical entanglement temperature for $\ell = 0.6$ (left), $\ell = 1$ (middle) and $\ell = 1.5$ (right) as function of time, all for the the NESS with $\chi = \sqrt{\mathcal{E}_C/\mathcal{E}_H} = 9/16$.

smaller region, in which case $T_{\text{ent}}^{\text{dyn}}$ can be non-monotonic. One reason is that $T_{\text{ent}}^{\text{dyn}}$ for $\ell = 0.6$ is larger at late times for the rarefaction wave, even though the physical temperature has decreased from the hot temperature towards the lower steady state temperature.

# 6  Discussion

To the best of our knowledge, our work is the first successful simulation of the dynamic formation of NESSs in a holographic field theory in four spacetime dimensions. We considered the evolution of energy and charge densities in particular, as well as the evolution of entanglement entropy.

Let us recall our main results. Most importantly, our holographic results are consistent with a solution involving a shock wave travelling towards the cold bath and a rarefaction wave travelling towards the hot bath, with a steady state region forming in between. The wave moving towards the cold side approaches a steep, but smooth wave with time independent profile and finite entropy production at late times. The wave moving towards the hot side is progressively broadening and approaches a rarefaction wave with zero entropy production at late times. At sufficiently late time the properties of the NESS region (energy density, charge density, etc) are numerically very close to those of an analytic shock+rarefaction wave solution (less close to a shock+shock solution) of the Riemann problem.

For the dynamics of a conserved $U(1)$ charge density, we find that there emerge two separate plateaus with different charge density inside the NESS region, as expected from the analogous Riemann problem. In contrast to the Riemann problem however, for which these plateaus are separated by a discontinuity, in our holographic simulation this transition region is realised as smooth crossover that broadens in time.

We also investigated the evolution of entanglement entropy of spatial sub-regions regions crossed by our holographic shock and rarefaction waves using the Hubeny-Rangamani-Takayanagi prescription [18] to determine the HEE in our time dependent setting. Subject to appropriate normalisation by which the energy density and HEE are chosen to agree in the NESS and thermal regions, the evolution of the entanglement entropy follows closely the

evolution of the energy density, except for a small time delay that is more pronounced for the rarefaction wave than for the shock wave. Inspired by the first law of entanglement, we define the dynamical entanglement temperature as the ratio of the entanglement entropy and the spatial integral over the energy density inside the entangling region.

There are many interesting future directions. A logical extension of the current work is to turn on the back-reaction of the gauge field to the metric [37]. Physically this implies that the pressure is not a function of the energy density only in the hydrodynamic regime, but also depends on the charge density. It would be interesting to see how features such as the contact discontinuity in the charge density change when the back reaction is taken into account. Including the back-reaction will also allow to study the relation of the null energy condition in the gravity dual to the dynamics of the stress tensor in the boundary theory along the lines of [56].

One could introducing a scalar field [57] with non-trivial potential to study the effect of conformal symmetry breaking on the properties of the NESS. Another possibility is to investigate the effect of finite coupling corrections with simulations in Gauss–Bonnet gravity [58, 59]. It would also be interesting to include transverse flow, in the presence of which not only the charge density, but also the energy density can develop a contact discontinuity [60]. Furthermore, it would be interesting to compare our results against solutions recently obtained from the equations of relativistic hydrodynamics of non-perfect fluids [61].

It would be very interesting to study the time evolution of HEE in membrane theory, using [52,54] as a starting point to study the long time and large scale dynamics of the entanglement entropy. In that case the extremal surface computation reduces to a minimisation problem that is much easier to study numerically with e.g. the Surface Evolver [62]. The current setting could be a perfect playground to employ membrane theory in non-homogeneous settings, as it would be possible to use the analytically available metric (43) in the shock region. Generally, such a study will provide new information on velocity bounds for the holographic shock+rarefaction wave solution.

Moreover, it will also be of interest to make contact with the analysis of [63] which studies NESS using a quasinormal mode approach, in a rather different setup though that involves a forced flow across obstacles in which inhomogeneous, but time-independent states form.

Finally, it is highly desirable to numerically probe the far-from-equilibrium dynamics of the system considered in this paper under more extreme conditions, for $\chi \ll 1$. Fig. 11 represents our efforts in this direction, but we found it prohibitively hard to numerically achieve stable time evolution at smaller ratios of the initial energy densities. The question whether the breakdown of our simulations at small $\chi$ is just an artefact of our numerical scheme or if it indicates a physical instability is currently not clear and deserves further investigation.

# Acknowledgements

It is our pleasure to thank Daniel Grumiller, Giuseppe Policastro and Luciano Rezzolla for comments on the manuscript and Koenraad Schalm for useful discussions in the early stages of this project.

# A  Rankine–Hugoniot jump conditions

In this appendix we review the derivation of the Rankine-Hugoniot jump conditions. We start with the following Riemann problem

$$\partial_t q(t,x) + \partial_x f(q(t,x)) = 0, \quad q(0,x) = \begin{cases} q_L & \forall x < 0 \\ q_R & \forall x > 0 \end{cases}, \tag{76}$$

where $q(t,x)$ is a conserved charge and $f(q(t,x))$ the associated flux. A discontinuous solution of (76) can be obtained from the integrated conservation law

$$\partial_t \left( \int_{x_L}^{x_S(t)} q(t,x)dx + \int_{x_S(t)}^{x_R} q(t,x)dx \right) = - \int_{x_L}^{x_R} \partial_x f(q(t,x)), \tag{77}$$

where $x_S(t)$ parametrizes the location of the discontinuity at time $t$ and the integration bounds are chosen such that $x_L < x_S(t) < x_R$. To simplify the expression on the left hand side we use Leibniz integral rule

$$\partial_t \left( \int_{a(t)}^{b(t)} g(t,x)dx \right) = g(t,b(t))\frac{d}{dt}b(t) - g(t,a(t))\frac{d}{dt}a(t) + \left( \int_{a(t)}^{b(t)} \partial_t g(t,x)dx \right). \tag{78}$$

This gives

$$q_L x'_S(t) + \lim_{\epsilon \to 0^+} \int_{x_L}^{x_S(t)-\epsilon} \partial_t q(t,x)dx - q_R x'_S(t) + \lim_{\epsilon \to 0^+} \int_{x_S(t)+\epsilon}^{x_R} \partial_t q(t,x)dx = -(f_R - f_L), \tag{79}$$

where we defined $f_{R/L} = f(q_{L/R})$. The remaining two integrals vanish when taking the limits $x_L \to x_S(t) - \epsilon$ and $x_R \to x_S(t) + \epsilon$. We arrive at the Rankine-Hugoniot jump condition

$$v_S(q_L - q_R) = f_L - f_R, \tag{80}$$

where $v_S = x'_S(t)$ is the propagation speed of the shock.

# B  Sensitivity to initial conditions and diffusion

Numerically it is difficult to initialize the evolution of our coupled heat baths with a truly discontinuous step function. In practice we approximate the discontinuous interface by a smooth function of the form $\tanh(cz)$, which in the limit $c \to \infty$ converges to the Heaviside theta function. In the following we verify that the evolution is insensitive to the choice of the constant $c$ that determines the steepness of the initial interface. For this we compare in Fig. 19 two evolutions, where solid lines equal the evolution in Fig. 5 and the dashed lines an evolution with an 1.5 times smaller value of $c$. At $t = 0$ the smaller value of $c$ gives a wider profile, but after a time $t \gtrsim 4$ the two simulations are virtually indistinguishable.

A related but separate problem is the set-up of two baths of charge at different chemical potential in a space at constant temperature. We simulated this situation by using the initial condition of Fig. 5, with the sole difference that we made both temperatures equal to the temperature of the cold bath. This problem is in particular relevant for the contact

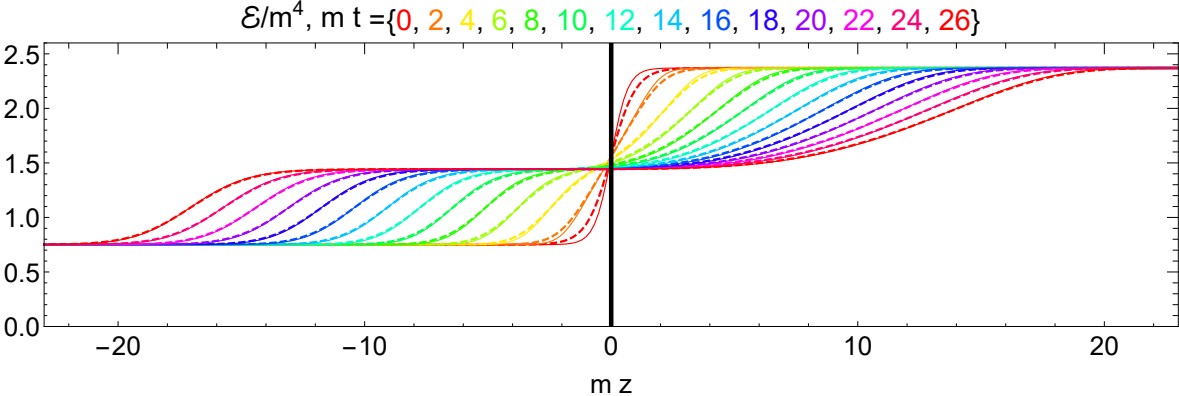

Figure 19: Evolution of the energy density for two different initial profiles: $c = \frac{3}{2}$ (solid) and $c = 1$ (dashed). For $mt \gtrsim 4$ the evolution is insensitive to the initial condition.

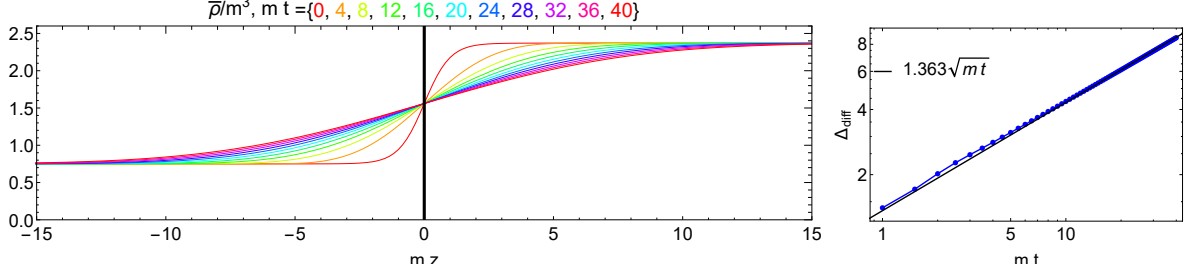

Figure 20: Left: Diffusion of the charge density in a heat bath with constant temperature. Right: Evolution of the charge diffusion width $\Delta_{\mathrm{diff}}(t)$. The blue line is the result extracted form the numeric simulation and the black line the analytic fit.

discontinuity that exists in the steady state region as presented in Sec. 2 and elaborated on in Sec. 5. The results are shown in Fig. 20. Indeed, we find that the diffusion of charge is qualitatively different as compared to the shock and rarefaction waves. To quantify the difference we define the (time dependent) charge diffusion width $\Delta_{\text{diff}}(t)$ as the spatial distance between the two points where the charge density is at 25% and 75% in between the two baths with lower and higher charge, respectively. On the right hand side of Fig. 20 we plot the time evolution of $\Delta_{\text{diff}}$ which clearly follows the expected $\sqrt{t}$ scaling.

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
