# Peer review of "Non-equilibrium steady state formation in 3+1 dimensions"

_SciPost Physics_

## Round 2 · Referee Report · Anonymous (Referee 1) · 2021-5-31

Report

The paper explores a Non-Equilibrium Steady State (NESS) in a four-dimensional theory with a gravity dual. The authors solve numerically Einstein's equations for the gravitational system and study the physical properties of the dual NESS, including the evolution of the entanglement entropy. When possible, the results are compared with the ones that can be extracted from the corresponding solution of the Riemann problem. The results are interesting, the study is complete and rigorous, so I will recommend it for publication. Nevertheless, I would like to pose a few questions/observations that might improve the presentation: 1) Do the authors have any idea or comment about the cause of the difference between the contact discontinuity for the charge density in the Riemann problem and the smooth interpolating configuration ("crossover region") they find in the gravity solution? Of course one cannot expect to see a sharp discontinuity in a numerical analysis starting from a smooth initial condition, but the behavior of the interpolating configuration does not seem to evolve to a discontinuity-like form, even at late time. 2) Would an analysis as the one in figure 12, but for the "crossover region", have been helpful in understanding its behavior? Maybe it would have suggested a possible analytic form. 3) Actually, in the original coordinates also the "shock wave" seems to be different from the one of the Riemann problem, since it expands with time. It is only in the z/t coordinates that it seems to evolve to a shock wave at late times. Why should we expect to see this behavior only in the z/t coordinates? 4) The backreaction of the charge on the metric has been ignored. Is it correct to state that this approximation does not allow for a check that the two regions of different charge density within the NESS do not correspond to two different energy densities? A-priori the metric could be influenced by these two charges and the energy density could be different.
5) At page 6, 3 lines after formula (12), it is stated that "For larger values of \chi the charge density becomes non-monotonic:", but from figure 1 the non-monotonicity seems to appear for smaller values of \chi, doesn't it? 6) In the formulas (38) and (42) for the temperature I believe there is a sign mistake, otherwise the chargeless limit of the solution would result in a negative temperature. 7) I would have liked to see a citation to some literature in appendix A.

Let me also point out a few typos: a) In the last line of the caption of figure 2 "hot" should be written as "Hot". b) I would have liked to read the last sentence of the paragraph after formula (43) somewhere around formula (41): it would have been immediately useful. c) In the sentence starting 4 lines after formula (51) there is a "are" too much. d) I believe the "red" in the 5th line of the caption of figure 9 should be a "blue". e) In formula (73) the is a dot missing. f) Formula (74) is the inverse of the wordy definition in the 2 lines above it, right? g) Formula (77) is lacking a "dx" on the r.h.s.

---

## Round 2 · Referee Report · Anonymous (Referee 2) · 2021-6-18

Report

The paper on “Non-equilibrium steady state formation in 3+1 dimensions” by Ecker, Erdmenger and van der Schee is to the best of my knowledge the first dynamical, fully non-linear study of what could be called the ‘UV completion’ of the hydrodynamic Riemann problem in 3+1 dimensions - a historically important example of a NESS. Needless to say, the study is holographic as no other method exists that could perform an even remotely comparable analysis in a complicated interacting quantum field theory.

The work is technically proficient, correct, very interesting and timely, as well as well written. As a result, I do not have any major objections to it being published almost immediately by SciPost. I only have one comment regarding the contents and a few smaller comments, potentially typos.

I think that when the second law of thermodynamics is discussed and its violations, it is important to state that this is the local version of the second law. The closest thing to a proof of this statement exists in 1612.07705, which is based on the Schwinger-Keldysh effective field theory of hydrodynamics developed in 1305.3670, 1405.3967, 1511.03646, 1511.07809, 1701.07436 and numerous other papers. My suggestion for the authors is to extend the discussion of this and comment further on the meaning of this breaking of the local second law in some cases and how this is restored in the full UV-complete simulation. This is naturally a very interesting aspect of their work and shows how important it is go beyond hydrodynamics. It would be nice to discuss and stress this further and well as discuss in greater depth what this law really means and why it is sometimes violated.

My smaller comments are as follows:

  • Regarding the second law, in the introduction, it says that the authors ‘confirm that the double-shock solution violates the second law of thermodynamics also in 3+1 dimensions’. What does it mean that they are confirming it? Was it predicted somewhere? In that case, they could add a reference. I suppose they are confirming a general expectation that similar things should happen in all dimensions.

  • The authors use words ‘analytic’ and ‘numeric’ several times. Should it not be ‘analytical’ and ‘numerical’, especially when it comes to the latter?

  • Above eq. (17) I think that it should be ‘monotonic’ not ‘monotonous’.

  • Below eq. (27), what does it mean that a ‘similar expression’ can be derived from the Rankine-Hugoniot condition. Why similar and not the same?

---

## Round 2 · Referee Report · Anonymous (Referee 3) · 2021-6-19

Strengths

1- extensive study both analytical and numerical 2- numerics agrees well with analytics 3- important subject 4- good explanations

Weaknesses

1- presentation of some numerical results not optimal 2-discussion of viscosity not clear enough

Report

In this paper the authors consider problem of non-equilibrium steady states in 3+1 dimensions. They study the result of an initial imbalance of charge and density in relativistic models. They use both conformal hydrodynamics and Einstein’s equations in order to represent N=4 super Yang-Mills. The results suggest that the standard solution to the Riemann problem correctly describes what is obtained numerically in the holographic representation.

The paper is interesting and gives an extensive discussion of this non-equilibrium problem. The problem was studied in the past, but never to this extent. What is particularly interesting is the verification against numerical solution of the holographic problem. Basically, this shows that conformal hydrodynamics is a good effective description of the equivalent non-equilibrium problem in gravity, and, less directly, further suggests that both are good description of N=4 SYM. Another interesting aspect is that entanglement entropy is also studied, albeit only in the holographic description.

I believe the paper should be published, after some minor clarifications have been made.

The Riemann problem and its solutions in the form of shocks and rarefaction waves, and the entropy conditions for physically meaningful solutions, are very well-known aspects of hyperbolic systems of equations like those of conformal hydrodynamics, see for instance the set of lecture notes

Hyperbolic Conservation Laws: An Illustrated Tutorial, https://link.springer.com/chapter/10.1007/978-3-642-32160-3_2

The discussion by the authors is correct and clear, but for full reference it would be good to also cite either the above or some other standard textbook on the subject.

Section 2:

It may be worth noting earlier in the discussion of shocks etc that the fact that the entropy current is broken at shocks is also well known, and relates to the fact that shocks are scaled solutions to the full viscous equations. Viscosity leads to non-negative entropy production by general theorems, thus solutions with negative entropy production cannot arise as Euler scaling of a physically meaningful viscous hydrodynamic equation.

Viscous hydrodynamics is not the main point of the paper, but it is mentioned in the abstract, introduction and then on page 22, yet it does not seem to be discussed properly in section 2. In particular, what equation exactly is referred to on page 22 and later? Viscosity is discussed in the holographic section on page 17, it is unclear to me how this relates to hydrodynamic viscosity.

It is mentioned that the back relation on gauge field is neglected (and, in the conclusion, that is a point to look at in the future). Can the authors comment on how important this assunption is? Is this related to assumption that pressure is independent of charge density?

How does the solution to the Riemann problem with gauge field discussed in Sect 2 compare with that of [20]?

Section 5:

Figure 6: it is not clear what times correspond to the curves for the shock+shock, shock+rarefaction, and ideal hydro solution; please clarify this.

Fig 8: I find this figure clearer than Fig 6, but it would be nice to have there too, or in a different but similar-style figure, the comparison with shock+shock and ideal hydro.

Ideal hydro: as I understand, this is simply the hydro evolution of an initial condition that “resembles” the initial condition used in holography. Could the authors confirm if the initial condition taken only resembles, or does it exactly reproduce, the initial profile of energy and charge density used for the holographic numerics? It would be nice to have it exactly reproducing the profiles of the numerics.

Fig 10 and discussion around gives a good analysis, using viscous hydro. Again, it would be good to be more precise: exact viscous hydro equation used, viscosity value, and initial condition agreeing exactly with that of numerics.

It may also be worth emphasising that the differences between shock+rarefaction and holographic is mainly due to the different (smooth) initial condition in the holographic numerics.

Requested changes

as per the report:

1- add general references on Riemann problem for hyperbolic equations, and some explanations for choosing the correct physical solution

2- add discussion of viscous hydro in early sections, in the context of hydrodynamics, and connect with that made in the context of holography and in the analysis of the results

3- discuss how solution to Riemann problem presented in Sect 2 compares with previous literature and general textbook material

4- Clarify Figs 6 and Fig 8

5- Clarify the "ideal hydro" solution and its relation with the holographic calculation

6- Clarify early enough in the discussion the origin of the bulk of the difference between shock+rarefaction and holographic solution

---

## Editorial Decision

resubmitted